# Comparison of ozone profiles and influences from the tertiary ozone maximum in the night-to-day ratio above Switzerland

Lorena Moreira[1], Klemens Hocke[1], and Niklaus Kämpfer[1]

[1]Institute of Applied Physics and Oeschger Centre for Climate Change Research, University of Bern, Bern, Switzerland

*Correspondence to:* L. Moreira (lorena.moreira@iap.unibe.ch)

**Abstract.** Stratospheric and middle mesospheric ozone profiles have been continually measured by the GROMOS (GROund-based Millimeter-wave Ozone Spectrometer) microwave radiometer since 1994 above Bern, Switzerland (46.95°N, 7.44°E, 577 m). GROMOS is part of the Network for the Detection of Atmospheric Composition Change (NDACC). A new version of the ozone profile retrievals has been developed with the aim to improve the altitude range of retrieval profiles. GROMOS profiles from this new retrieval version have been compared to coincident ozone profiles obtained by the satellite limb sounder Aura MLS. The study covers the stratosphere and middle mesosphere from 50 to 0.05 hPa (from 21 to 70 km) and extends over the period from July 2009 to November 2016, which results in more than 2800 coincident profiles available for the comparison. On average, GROMOS and MLS comparisons show agreement generally over 20% in the lower stratosphere and within 2% in the middle and upper stratosphere for both daytime and nighttime, whereas in the mesosphere the mean relative difference is below 40% at daytime and below 15% at nighttime. In addition, we have observed the annual variation of nighttime ozone in the middle mesosphere, at 0.05 hPa (70 km), characterised by the enhancement of ozone during wintertime for both ground-based and space-based measurements. This behaviour is related to the middle mesospheric maximum of ozone (MMM).

## 1 Introduction

Passive millimeter wave radiometry is a well-established technique to monitor atmospheric constituents by detecting the radiation emitted by the rotational transitions of the molecules. It makes use of the spectral properties of the atmospheric species in order to derive information about their distribution in the atmosphere. The main advantages of this technique are its independence of solar irradiation and its insensitivity to weather conditions and aerosols. Additionally it offers a good temporal resolution of 1 hour. Measurements of ozone performed by this technique have been indispensable in monitoring changes in the ozone layer and improving the comprehension of the processes that control ozone abundances (e.g. Steinbrecht et al., 2009). Stratospheric ozone, in spite of its small abundance, plays a beneficial role by absorbing most of the biologically harmful ultraviolet sunlight. The absorption of UV radiation by ozone creates a source of heat, therefore ozone plays a key role in the temperature structure of the Earth's atmosphere. Changes in the stratospheric ozone concentration alter the radiative balance of the atmosphere, the atmospheric composition and the dynamics of the atmosphere. Continuous long-term monitoring of ozone is essential for the detection of long-term trends of the stratospheric ozone layer (e.g. WMO, 2014). The ground-based ozone radiometer GROMOS (GROund-based Millimeter-wave Ozone Spectrometer) is part of the Network for the Detection

of Atmospheric Composition Change (NDACC). In order to satisfy the requirements of accuracy and stability the validation of instruments is necessary. There have been a number of comparisons in the past, showing that GROMOS is a reliable tool to measure stratospheric and lower mesospheric ozone (WMO, 2014; Studer et al., 2013; van Gijsel et al., 2010; Keckhut et al., 2010; Dumitru et al., 2006).

This manuscript presents a comparison between the data from the ground-based instrument GROMOS and the space-based instrument MLS for the time interval from July 2009 to November 2016 covering the stratosphere and the middle mesosphere, which corresponds to the altitude range from 20 to 70 km (50 to 0.05 hPa). We have also performed an analysis of the diurnal variation and its amplitude (night-to-day ratio) of middle mesospheric ozone, at 0.05 hPa (70 km). The diurnal variation of ozone in the lower and middle mesosphere is observed as an increase in ozone after sunset and a decrease after sunrise.

Daytime production of atomic oxygen by photolysis of ozone (Reaction R7) and photolysis of molecular oxygen (Reaction R5) results in nighttime ozone production by recombination of atomic and molecular oxygen (Reaction R6) (Brasseur and Solomon, 2005). In addition, we observe the annual variation of the nighttime mesospheric ozone with a maximum in wintertime and a minimum in summertime. This maximum of mesospheric ozone during nighttime in winter is related to the middle mesospheric maximum of ozone (MMM) (e.g., Sonnemann et al., 2007; Hartogh et al., 2004) also known as the tertiary ozone

maximum (e.g., Sofieva et al., 2009; Degenstein et al., 2005; Marsh et al., 2001). Sonnemann et al. (2007) reported that the MMM is a phenomenon that occurs at high latitudes close to the polar night terminator around 72 km altitude during nighttime in winter and extends into middle latitudes with decreasing amplitude. Marsh et al. (2001) interpreted the tertiary peak by considering that in the middle mesosphere during winter, with solar zenith angle close to $90°$, the atmosphere becomes optically thick to UV radiation at wavelengths below 185 nm and, since photolysis of water vapour (Reaction R1) is the primary source

of odd-hydrogen, reduced UV radiation results in less odd-hydrogen. The lack of odd-hydrogen needed for the catalytic depletion of odd-oxygen (Reactions R2, R3 and R4), in conjunction with an unchanged rate of odd oxygen production (Reaction R5), leads to an increase in odd-oxygen. This results in higher ozone concentration because atomic oxygen recombination (Reaction R6) remains as a significant source of ozone in the mesosphere. Additionally, Hartogh et al. (2004) extended the interpretation by considering the very slow decrease of the ozone dissociation (Reaction R7) rate with increasing solar zenith

angle.

$$H_2O + h\nu(\lambda < 185nm) \rightarrow OH + O \tag{R1}$$

$$O + OH \rightarrow O_2 + H \tag{R2}$$

$$H + O_2 + M \rightarrow HO_2 + M \tag{R3}$$

$$O + HO_2 \rightarrow O_2 + OH \tag{R4}$$

$$O_2 + h\nu(\lambda < 242\text{nm}) \rightarrow O + O \tag{R5}$$

$$O + O_2 + M \rightarrow O_3 + M \tag{R6}$$

$$O_3 + h\nu \rightarrow O_2 + O \tag{R7}$$

The next section describes briefly both instruments and measurement techniques. The results of the comparison are shown in Section 3. Section 4 analyses the night-to-day variability and provides a short discussion, and the conclusions are summarised in Section 5.

## 2 Instruments and measurement techniques

### 2.1 The ground-based microwave radiometer GROMOS

This study is based on stratospheric and mesospheric ozone volume mixing ratio (VMR) profiles observed by GROMOS. The ground-based millimeter wave ozone spectrometer has been operating in Bern, Switzerland (46.95°N, 7.44°E, 577 m) since November 1994 in the framework of the Network for the Detection of Atmospheric Composition Change (NDACC). The in-
15 strument measures the thermal microwave emission of the pressure broadened rotational transition of ozone at 142.175 GHz. The vertical distribution of ozone VMR can be retrieved from the measured spectral line since it contains information on the altitude distribution of the emitting molecule due to the pressure broadening. The retrieval procedure is performed through the Atmospheric Radiative Transfer Simulator (ARTS2) (Eriksson et al., 2011) which is used as a forward model to simulate the atmospheric radiative transfer in a modelled atmosphere and so calculate the ozone spectrum of this modelled atmosphere. The
20 a priori profile of $O_3$ VMR required for the retrieval is taken from a monthly varying climatology from ECMWF reanalysis until available (70 km) and extended above by an Aura MLS climatology (2004 to 2011). The line shape used in the retrieval is the representation of the Voigt line profile from Kuntz (1997). Spectroscopic parameters to calculate the ozone absorption coefficients were taken from the JPL catalogue (Pickett et al., 1998) and the HITRAN spectroscopic database (Rothman et al., 1998). The atmospheric temperature and pressure profiles are taken from the 6 hourly of the European Centre for Medium-
25 Range Weather Forecast (ECMWF) operational analysis data and are extended above 80 km by monthly mean temperatures of the CIRA-86 Atmosphere Model (Fleming et al., 1990). The accompanying Matlab package Qpack2 (Eriksson et al., 2005) compares the modelled spectrum with the measured spectrum and derives the best estimate of the vertical profile by using the optimal estimation method (OEM) (Rodgers, 1976). The OEM also provides a characterisation and formal analysis of the uncertainties (Rodgers, 1990).

Recently, we have developed a new retrieval version (version 150) with the aim to optimise the averaging kernels. The differences with the former version (version 2021) are in the a priori covariance matrix, in the measurement error and in the integration time of the retrieval.

In version 2021 the diagonal elements of the a priori covariance matrix are variable relative errors ranging from 35% at 100 hPa to 28% in the lower stratosphere and increasing with altitude from 35% in the upper stratosphere up to 70% in the mesosphere. Meanwhile, in version 150 the a priori covariance matrix has a constant value for the diagonal elements of 2 ppm. For both retrieval versions the off-diagonal elements of the a priori covariance matrix exponentially decrease with a correlation length of 3 km.

Regarding the measurement noise, in version 2021 it is a constant error of 0.8 K whereas in version 150 we used a variable error depending on the tropospheric transmission:

$$\Delta T'_b = 0.5 + \frac{\Delta T_b}{e^{-\tau}} \tag{1}$$

the error of the measured brightness temperature, $\Delta T_b$, is given by the radiometer equation:

$$\Delta T_b = \frac{T_b + T_{rec}}{\sqrt{\Delta f \cdot t_{int}}} \tag{2}$$

The radiometer equation gives the resolution of the radiation measured, which is determined by the bandwidth of the individual spectrometer channels ($\Delta f$), by the integration time ($t_{int}$) and by the total power measured by the spectrometer. A constant error of 0.5 K is considered as a systematic bias of the spectra, due to spectroscopic errors and the water vapour continuum. The error of the brightness temperature ($\Delta T_b$) is of the order of a few Kelvins in the line centre and 0.5 K in the line wings of the spectrum. Therefore the measurement noise ($\Delta T'_b$) depends on the bandwidth of the spectrum and on the tropospheric transmittance. This is a more realistic approach for the retrieval than considering a constant measurement noise, resulting in an improvement in the retrieved ozone VMR in the lower stratosphere. The sampling time for version 150 is 1 hour and in case of version 2021 is 30 minutes. Longer integration time improves the retrieved ozone VMR at upper altitudes.

In Figure 1 is displayed a comparison between version 2021 and version 150 of ozone profiles measured by GROMOS for the time interval from July 2009 to November 2016. In the left panel are represented the mean ozone profiles retrieved by version 2021, in red, and by version 150, in blue. The standard deviation of the ozone VMR are shown by the coloured areas, red in case of v2021 and blue for v150. The mean relative differences (blue line in the middle panel) and the volume mixing ratio (VMR) differences (blue line in the right panel) are ranging from 30% (0.5 ppm) in the lowermost stratosphere to within 5% (0.2 ppm) in the middle stratosphere, and increasing to 10% (0.4 ppm) in the upper stratosphere and up to 18% (0.05 ppm) at 0.05 hPa (70 km). The blue areas in the middle and right panels represent the standard deviation of the differences, relative differences and VMR differences, respectively. We can conclude from Figure 1 that the differences between version 2021 and version 150 appear in the lower stratosphere and in the mesosphere.

Figure 2 displays an example of a GROMOS retrieval accomplished by the new retrieval version 150. The left panel show the a priori (green line) and the retrieved profile (blue line) measured in July 2013 at noon. In the middle panel are represented the averaging kernels (AVK) and the area of the averaging kernels (measurement response). The AVKs are multiplied by 4 in order

to be displayed along with the measurement response (red line). The AVK-lines are grey except for some selected altitudes, which are shown in different colours to make the Figure 2 easier to interpret. AVKs are a representation of the weighting of information content of the retrieval parameters therefore an estimate of the a priori contribution to the retrieval can be obtained by 1 minus the area of the AVK (measurement response). It is considered a reliable altitude range of the retrieval when the true state dominates over the a priori information, i.e. where the measurement response is larger than 0.8 (an a priori contribution smaller than 20%). The measurement response shown in Figure 2 is around 1 from 18 to 70 km. The magenta line in the right panel shows the altitude peak of the corresponding kernels and proves that the AVK peak at its nominal altitude for the considered altitude range. And finally, the cyan line displays the vertical resolution which is quantified by the full width at half maximum of the averaging kernels. The vertical resolution of this new retrieval version of GROMOS lies from 10 to 15 km below 40 km altitude and from 15 to 20 km below 70 km altitude. In version 2021, the vertical resolution lies generally within 10–15 km in the stratosphere and increases with altitude to 20–25 km in the lower mesosphere. Between 20 to 52 km (50 to 0.5 hPa) the measurement response is higher than 0.8. For more details on version 2021 we refer to Moreira et al. (2015). Comparing the measurement response and the vertical resolution obtained by version 2021 and by version 150 we can conclude an improvement in the results retrieved by version 150. We assume that the changes performed in the a priori covariance matrix, in the measurement noise and in the integration time result in the improvement of the retrieval product, mainly observed in the lowermost and in the uppermost limit of the retrieved ozone VMR profile.

For technical details, measurement principle of the instrument, see for example Moreira et al. (2015) and Peter (1997) and references included therein.

## 2.2 The Aura Microwave Limb Sounder

The Microwave Limb Sounder (MLS) is a passive microwave limb-sounding radiometer onboard the NASA Aura satellite. The Aura spacecraft was launched in 2004 into a near polar, sun-synchronous orbit with a period of approximately 100 minutes. The satellite overpasses the GROMOS measurement location (at northern midlatitudes) twice a day, approximately around noon and midnight. The standard product for ozone is derived from MLS radiance measurements near 240 GHz. The vertical resolution of the ozone profiles ranges from 3 km in the stratosphere to 6 km in the mesosphere (Schwartz et al., 2008). The present study has used ozone profiles from version 4.2. A summary of the quality of version 4.2 Aura MLS Level 2 data can be found in Livesey et al. (2016). Details about the Aura mission can be found in Waters et al. (2006).

## 3 Comparison of MLS and GROMOS

The vertical resolution of the MLS is within 3.5 km in the stratosphere and up to 5.5 km in the middle mesosphere. Therefore in order to compare ozone profiles of GROMOS with MLS, an averaging kernel smoothing is applied to the ozone profiles of the satellite data. The smoothed profile of MLS adjusted to the vertical resolution of GROMOS is expressed as:

$$\mathbf{x}_{\text{MLS,low}} = \mathbf{x}_{\text{a,GROMOS}} + \mathbf{AVK}_{\text{GROMOS}} \cdot (\mathbf{x}_{\text{MLS,high}} - \mathbf{x}_{\text{a,GROMOS}}) \tag{3}$$

being $\mathbf{AVK}_{\text{GROMOS}}$ is the averaging kernel matrix of GROMOS, $\mathbf{x}_{\text{MLS,high}}$ is the measured MLS profile and $\mathbf{x}_{\text{a,GROMOS}}$ is the a priori profile used during the retrieval procedure of GROMOS. The application of averaging kernel smoothing for the comparison of profiles with different altitude resolutions has been introduced and described by e.g. Tsou et al. (1995).

Every profile utilised in the comparison between MLS and GROMOS should be coincident in time and space. The requirement of time coincidence is satisfied when both measurements are within 1 hour in time. The selected criterion for spatial coincidence is that horizontal distances between the sounding volumes of the satellite and the ground station have to be smaller than $1°$ in latitude and $8°$ in longitude.

The present study extends over the period from July 2009 to November 2016 and covers the stratosphere and middle mesosphere from 50 to 0.05 hPa (from 21 to 70 km), and according to the spatial and temporal criteria, more than 2800 coincident profiles are available for the comparison. Figure 3a and Figure 3b show the mean ozone profiles of the collocated and coincident measurements of GROMOS (blue line), MLS convolved (red line) and MLS original (green line) at daytime and nighttime, respectively. The relative difference profile in percent given by $(\mathbf{x}_{\text{MLS,low}} - \mathbf{x}_{\text{GROMOS}})/\mathbf{x}_{\text{GROMOS}}$ is displayed in the middle panel of both Figure 3a and Figure 3b along with the standard deviation of the differences (blue area). The green line delimits the $\pm$ 10% area. The mean profile of the VMR differences is shown in the right panel of both Figure 3. The mean relative differences and the VMR differences at daytime (nighttime) are over 20% or 0.5 ppm (15% or 0.4 ppm) in the lower stratosphere and decreasing with altitude up to 0.7% or 0.02 ppm (2% or 0.06 ppm) at the stratopause and increasing with altitude up to 38% or 0.085 ppm (15% or 0.12 ppm) at 0.05 hPa (70 km). We conclude from Figure 3 that during nighttime GROMOS measures more $O_3$ VMR (ppm) than MLS except for the lower stratosphere, where MLS measures more $O_3$ VMR (ppm) than GROMOS, both at daytime and nighttime. Nevertheless in the mesosphere GROMOS measures more $O_3$ VMR (ppm) than MLS, both at daytime and nighttime.

For an overview on the differences between coincident profiles, the average over daytime and nighttime values of the ozone VMR (ppm) time series of GROMOS (blue line) and MLS (red line) are displayed in Figure 4 for different pressure levels. All time series have been smoothed by a moving average over 7 data points ($\sim$ 1 week). The agreement between both ground-based and satellite-based instruments depends upon altitude and time. A negative deviation of GROMOS series with respect to MLS occurs in the lower stratosphere. On the other hand, a positive deviation of GROMOS with respect to MLS is observed in the middle stratosphere for summers 2011, 2012, 2014 and 2015. Further, we notice a negative bias of GROMOS during summer 2016 from the stratopause towards the mesosphere. In Figure 5 are shown the scatter plots of averaged daytime and nighttime $O_3$ VMR measurements of GROMOS and MLS at the same pressure levels as Figure 4. The black lines, linear regression lines of the observations, are close to the green one to one lines, $O_3(\text{MLS})=O_3(\text{GROMOS})$, except for the lower stratosphere where we find the negative deviation of GROMOS with respect to MLS. The linear fit deviates from the identity where there is less ozone in the case of GROMOS during winter in the middle to upper stratosphere as we also observe in Figure 4, along with the positive deviation of GROMOS with respect to MLS during some summers. The calculation of the correlation coefficients also reveals good agreement with $r > 0.75$ for all altitudes levels except for the altitude above 50 km where $r$ is around 0.55.

## 4 Analysis of the night-to-day ratio

The diurnal variation of mesospheric ozone is characterised by an increase at the beginning of the nighttime and by a decrease after sunrise. This effect is explained by the recombination of atomic and molecular oxygen (e.g., Brasseur and Solomon, 2005). Because the ozone distribution in the mesosphere is mainly controlled by photochemistry, it depends strongly on the
5 solar zenith angle (Nagahama et al., 2003), therefore an annual variation is also expected in mesospheric ozone. Figure 6 shows both the diurnal variation of mesospheric ozone and the annual variation of nighttime mesospheric ozone. To analyse the variability of mesospheric ozone we have used ozone VMR measurements coincident in space and in time recorded by GROMOS and by MLS for the time period from July 2009 to November 2016. The first panel of Figure 6 displays the $O_3$ VMR measured at noon (GROMOS in red, MLS convolved in orange and MLS original in magenta) and at midnight (GROMOS
in blue, MLS convolved in cyan and MLS original in black) at 0.05 hPa (70 km) for the already mentioned time period. The original MLS data, i.e. not weighted with GROMOS AVKs, is shown in order to provide an insight of the observability of the effect of MMM at northern midlatitudes by GROMOS. We define as midnight (noon) value the average between the values recorded within 2 hours around midnight (noon). The daytime mesospheric ozone does not show any distinct annual variation. On the other hand, the annual variation of nighttime mesospheric ozone is characterised by a maximum in wintertime and a
minimum in summertime. The second panel of Figure 6 shows the evolution of the midnight mesospheric ozone throughout the year averaged for the time interval from July 2009 to November 2016. All time series displayed in both panels of Figure 6 have been smoothed in time by a moving average over 15 data points (∼1 week). A closer observation shows that the annual variation of the nighttime ozone exhibits a primary maximum over wintertime and a secondary maximum around springtime. Our results on the annual variation of mesospheric ozone at Bern (Switzerland, 46.95°N, 7.44°E) are in agreement with the
ones observed at Lindau (Germany, 51.66°N, 10.13°E) by Sonnemann et al. (2007). Disagreements appear in the amplitudes where the maximum values of GROMOS and MLS original do not exceed 1.5 ppm, 1.2 ppm in the case of MLS convolved, whereas at Lindau the maximum values exceed 3 ppm at 70 km. Nevertheless, our results are expected since this maximum of mesospheric ozone during nighttime in winter is related to the middle mesospheric maximum of ozone (MMM) and according to Sonnemann et al. (2007) its effect extends into midlatitudes with decreasing amplitude. Furthermore, we have analysed the
amplitude of the diurnal variation, the night-to-day-ratio (NDR). The NDR is closely related to the MMM, but it is also related to the change of the diurnal variation from winter to summer (Sonnemann et al., 2007). The annual variation of the NDR is modulated by oscillations of planetary time scale (Sonnemann et al., 2007). Sofieva et al. (2009) reported that during a sudden stratospheric warming event the tertiary ozone maximum can decrease significantly or can even be completely destroyed. Hocke (2017) has shown the loss of the tertiary ozone layer in the polar mesosphere due to the solar proton event in November
2004.

The first panel of Figure 7 displays the NDR of GROMOS (blue line) and MLS (red line) at 0.05 hPa (70 km) for the time interval from July 2009 to November 2016 while the second panel shows its evolution throughout the year averaged for the time interval under assessment. Both time series were smoothed in time by a moving average over 30 points (∼1 month). The orange line (MLS) and the cyan line (GROMOS) depicted in the second panel of Figure 7 show a moving average over 7 data points

(∼1 week) with the aim to clarify the understanding of Figure 7. Both the ground-based and the satellite-based instruments confirm the expected winter enhancement of the NDR, also observed at Lindau by Sonnemann et al. (2007), although the latter data exhibit larger amplitudes. We observe winter-to-summer values of a factor of one to two, whereas at Lindau, winter-to-summer values vary by a factor of 2–3 at 70 km (Sonnemann et al., 2007). Thus, despite the definition of the MMM being restricted to high latitudes, we can report its observation with a smaller amplitude at mid-latitudes.

## 5 Conclusions

Stratospheric and middle mesospheric ozone profiles for the period from July 2009 to November 2016 recorded by the ground-based instrument GROMOS and by the space-based instrument MLS were used to perform a comparison and to evaluate the diurnal variability and its amplitude, night-to-day ratio (NDR). The agreement between measurements coincident in space and time for both data records is within 2% (0.06 ppm) between 30 and 50 km (15–0.7 hPa) increasing up to 20% (0.5 ppm) at 20 km (50 hPa), for both daytime and nighttime. In the mesosphere the difference increases up to 38% (0.085 ppm) at daytime and up to 15% (0.12 ppm) at nighttime at 70 km (0.05 hPa). In general terms, we report good agreement between the new retrieval version (v150) of GROMOS and the version 4.2 of MLS. Furthermore, we observe extensions of the middle mesospheric maximum of ozone (MMM) during winter towards northern mid-latitudes. This effect is smaller in amplitude at mid-latitudes compared to high latitudes. Moreover, the winter enhancement of nighttime mesospheric ozone is observed by GROMOS and MLS above Bern.

## 6 Code availability

Routines for data analysis are available upon request by Lorena Moreira.

## 7 Data availability

The data from the GROMOS microwave radiometer is available via http://ftp.cpc.ncep.noaa.gov/ndacc/station/bern/hdf/mwave. MLS v4.2 data are available from the NASA Goddard Space Flight Center Earth Sciences Data and Information Services Center (GES DISC), http://disc.sci.gsfc.nasa.gov/Aura/data-holdings/MLS/index.shtml.

*Author contributions.* Klemens Hocke performed the retrieval of the GROMOS measurements. Lorena Moreira carried out the data analysis and prepared the manuscript. Niklaus Kämpfer is the principal investigator of the radiometry project. All authors have contributed to the interpretation of the results.

*Competing interests.* All authors declare that there are no conflicts of interest in the current version of the manuscript.

*Acknowledgements.* This work was supported by the Swiss National Science Foundation under Grant 200020 - 160048 and MeteoSwiss GAW Project: "Fundamental GAW parameters measured by microwave radiometry".

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

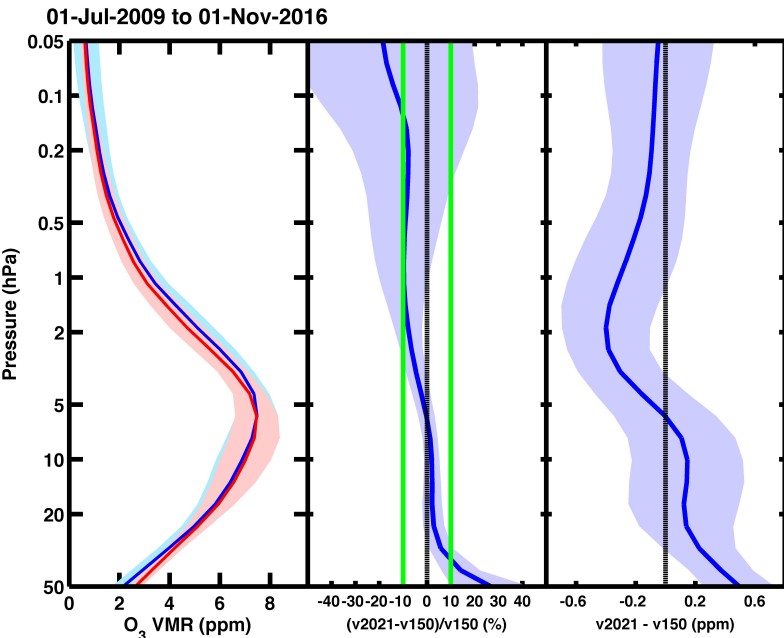

**Figure 1.** Mean ozone profiles retrieved by version 2021 (red line in the left panel) and by version 150 (blue line in the left panel) measured by GROMOS during the period from July 2009 to November 2016. The blue area (v150) and the red area (v2021) are the standard deviations of the ozone VMR. The mean relative difference profile (blue line) and the standard deviation of the differences (blue area) are represented in the middle panel, using the new version as reference. The green line delimits the ±10% area. In the right panel is shown the VMR difference profile along with its standard deviation

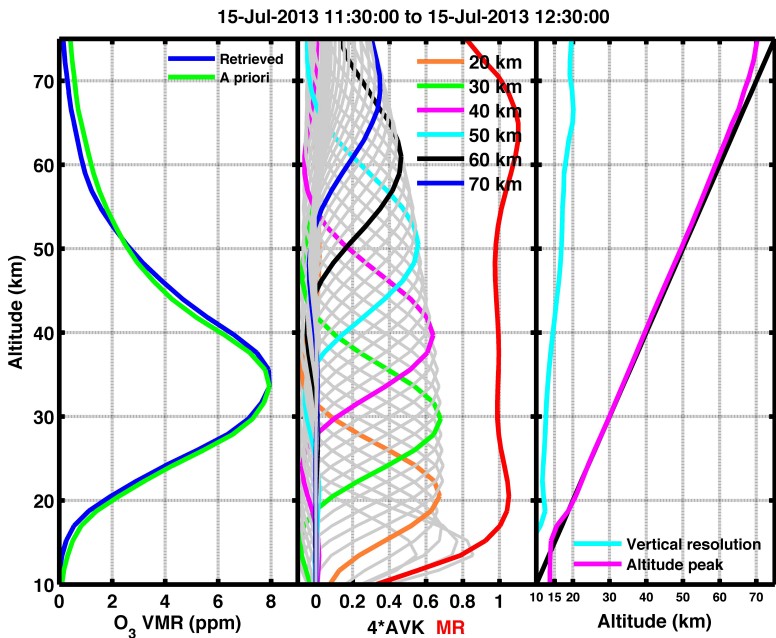

**Figure 2.** Example of an a priori profile and a retrieved ozone profile (green and blue lines in the left panel, respectively), averaging kernels (grey and colour lines in the middle panel), the measurement response (red line in the panel), vertical resolution (cyan line in the right panel) and altitude peak (magenta line in the right panel) of the GROMOS retrieval version 150 for July 15, 2013 with an integration time of 1 hour

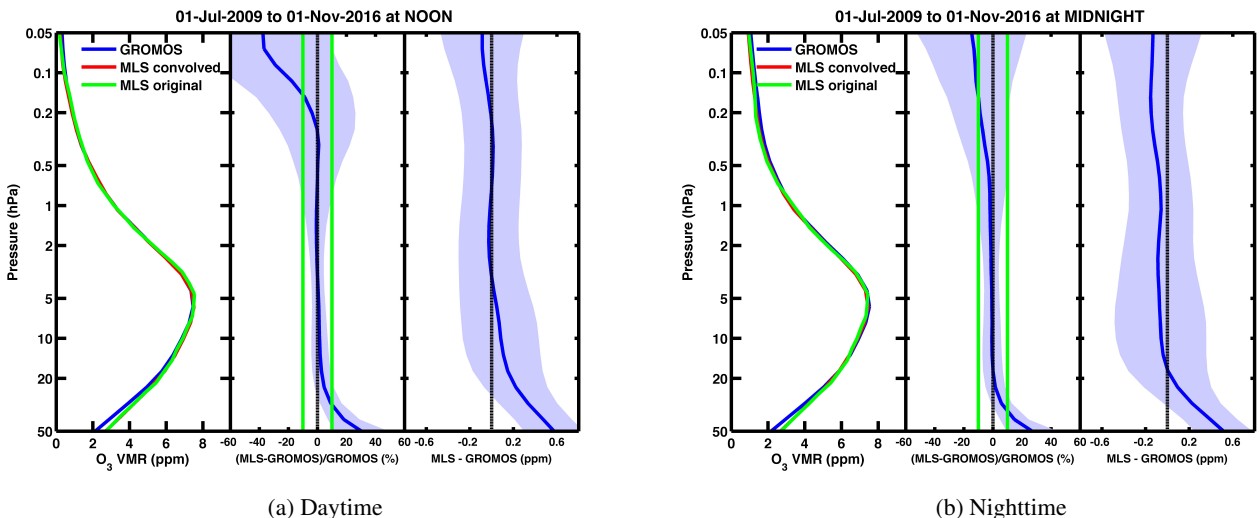

**Figure 3.** Mean ozone profiles recorded by GROMOS (blue line), MLS convolved (red line) and MLS original (green line) for the time interval between July 2009 and November 2016 are shown in the left panels of both daytime and nighttime Figures. The blue area (GROMOS) and the red area (MLS) are the standard deviations of the coincident measurements. The middle panels show the mean relative difference profile between data of both instruments, GROMOS as reference. The blue areas in the middle panels represent the standard deviation of the differences. The green lines in the middle panel delimit the ± 10% area. The mean VMR difference profile and its standard deviation (blue area) are displayed in the right panels of both daytime and nighttime, Figure 3a and Figure 3b, respectively

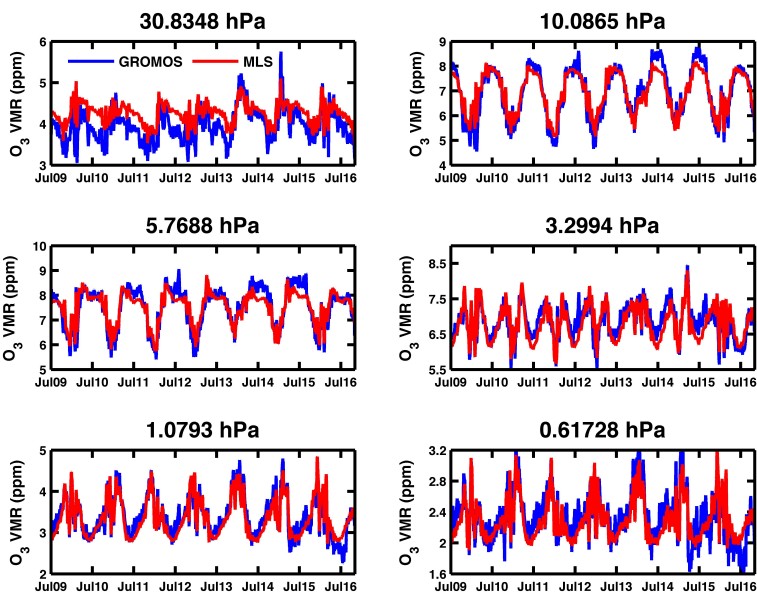

**Figure 4.** Time series of averaged daytime and nighttime $O_3$ VMR measurements of GROMOS (blue line) and MLS (red line) for the period from July 2009 to November 2016 at different pressure levels. An averaging kernel smoothing has been applied to the series of the MLS measurements coincident in time and space with the GROMOS measurements. Both time series are smoothed over 7 points or $\sim$1 week in time by a moving average

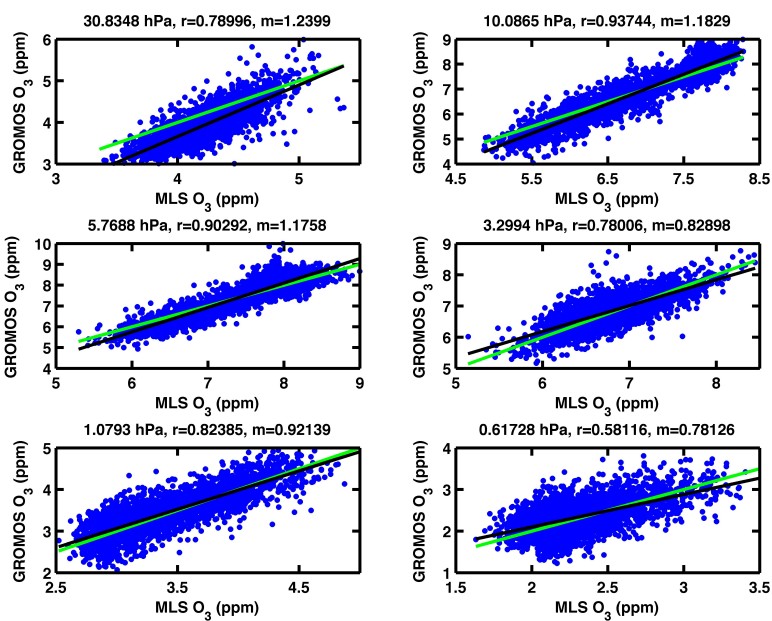

**Figure 5.** Scatter plots of coincident $O_3$ VMR measurements of GROMOS and MLS for the period from July 2009 to November 2016 at different pressure levels. The black line is the linear fit of both time series, and $m$ the slope of the linear fit. The green line indicates the case of identity, $O_3$(MLS)=$O_3$(GROMOS). $r$ values are correlation coefficients of the MLS and GROMOS time series

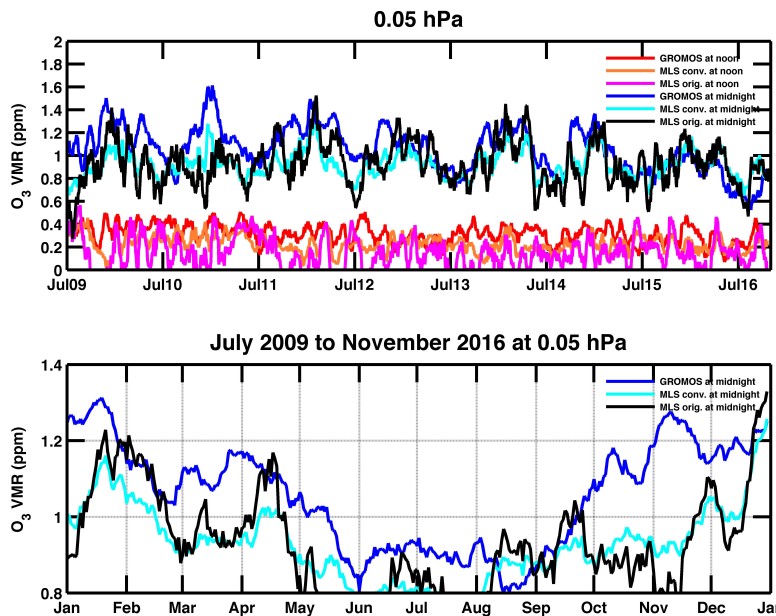

**Figure 6.** The first panel shows the diurnal variation of $O_3$ VMR measured at noon (GROMOS in red, MLS convolved in orange and MLS original in magenta) and at midnight (GROMOS in blue, MLS convolved in cyan and MLS original in black) at 0.05 hPa (70 km) and the second panel shows its evolution throughout the year averaged for the time interval under assessment (July 2009–November 2016). All time series are smoothed in time by a moving average over 15 points ($\sim$1 week)

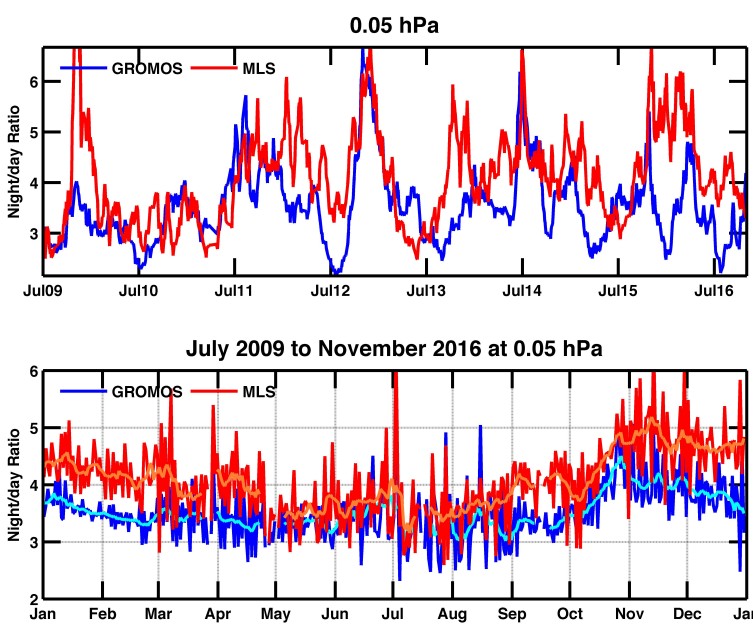

**Figure 7.** The first panel displays the night-to-day ratio (NDR) of GROMOS (blue line) and MLS (red line) at 0.05 hPa (70 km) for the time period from July 2009 to November 2016 and the second panel shows its evolution throughout the year averaged for this time period. The time series presented in the top panel are smoothed in time by a moving average over 30 data points (∼1 month) and the orange line (MLS) and the cyan line (GROMOS) shown in the second panel are averaged over 7 data points (∼1 week)