# Peer review of "Comparison of ozone profiles and influences from the tertiary ozone maximum in the night-to-day ratio above Switzerland"

_Atmospheric Chemistry and Physics, 2017_

## Referee Comment (RC1) · Anonymous Referee #1 · 2 May 2017

General Referee Comments:

This manuscript (by Moreira, Hocke, and Kampfer) describes comparisons of ozone profiles above Bern, Switzerland, between a ground-based millimeter-wave ozone spectrometer (GROMOS) and space-based data from the Aura Microwave Limb Sounder (MLS). The somewhat minor atmospheric discussions (that might justify submitting this work to ACP rather than AMT, for example) deal with the tertiary peak in ozone in (and above) the mesosphere as well as mesospheric diurnal and seasonal changes.

My main criticism has to do with the fact that there is not too much new regarding the upper atmosphere (e.g., Sonneman et al., 2007, and refs. therein mention the mid-

dle mesospheric maximum in ozone and its extension to mid-latitudes), and that the comparisons are not performed with too much attention to potential explanations of differences versus the satellite data near 0.05 hPa, which is near the upper range for both instruments, and in a region where the measurement response starts to weaken. If one wants to understand the absolute differences as well as differences in the seasonal variations, more care should be taken to clearly demonstrate the sensitivity to the profiles both above and below 0.05 hPa, as this region is close to a minimum in ozone, and can be quite sensitive to the profile and its a priori, in particular for pressures in the 0.02 to 0.005 hPa region. This is of concern when there are week-long periods during which the MLS and GROMOS tendencies vary in opposite ways (see Fig. 6), even if the annual cycles agree in a broad, semi-quantitative sense. It would be much better to at least give potential reasons, backed up with some quantitative analyses, especially since this is one of the main reasons for this manuscript (otherwise, one can refer to the work by Moreira et al., 2015, which was focused more on trends and limited to the more "valid" vertical range below 55-60 km). There should be a more complete discussion of how things are different in the new plots of averaging Kernels and acceptable vertical range, or one is left wondering whether the vertical range limit of 0.05 hPa is actually too optimistic (the recommended range in Moreira et al., 2005 stops at about 0.3 hPa, after all). What is new, and how does the newer retrieval really compare to the older one? This is discussed only briefly, but with essentially no comparison or detailed discussion.

Furthermore, Fig. 2 implies that MLS ozone values tend to be larger than GROMOS retrievals for pressures less than about 0.2 hPa (and larger by more than 50% near 0.05 hPa), wheras Fig. 5 shows GROMOS values usually larger than MLS values at 0.05 hPa (both day and night). This really would need to be clarified, besides the sensitivity tests that I am suggesting, given how sensitive to the region 10 km above and below the target altitude the GROMOS retrievals will be. Also, the MLS retrievals are not recommended for pressures less than 0.02 hPa; therefore, there is some sensitivity to MLS a priori values (for pressures less than 0.02 hPa) for the convolution

of MLS profiles that attempt to simulate the GROMOS vertical smoothing. If the authors feel that these more detailed analyses are somehow "beyond the scope" of this paper, then this remains too qualitative a study, in my view, and probably not worth publishing (essentially as is) in ACP; I would suggest shifting this to AMT in this case, which does not mean that somewhat better explanations for the differences should not be attempted. Finally, other datasets, such as MIPAS or ACE-FTS could be useful in helping to determine whether or not the absolute values and variations implied by GROMOS are sufficiently robust - or whether there are some significant issues with some of the satellite datasets. Such a study would be much more useful (wherever it gets published). I find that this work, at the very least, requires substantial clarifications (in addition to more relevant references regarding MLS data).

More specifics:

The ozone profile has a minimum in the region where the manuscript attempts to focus the reader's attention (near 0.05 hPa). Above and below this, there are strong gradients (with somewhat smaller values above, and then much larger values for pressures less than 0.01 hPa, especially during the night, and increasing values as well for lower altitudes into the upper stratosphere and lower mesosphere). The low resolution GROMOS profiles will depend on quite a range of values (within about 10-15 km of 0.05 hPa, above and below). The convolved results for MLS profiles (for comparison to GROMOS) also are taken over a limited time (very short compared to the observation period for GROMOS, which is at least an hour or two). Given that the lifetime of ozone is short in the middle mesosphere, there is also no discussion of the impact of the temporal sampling (or averaging) on these measurement comparisons. Obtaining larger values near 0.05 hPa for MLS (Fig. 2), if that has the right sign, could come from a priori values for pressures near or less than 0.01 hPa that are too large. Another potential explanation could be that GROMOS really does not have enough sensitivity at the uppermost altitudes and may therefore not sense the larger values well enough. Sensitivity tests or retrieval simulations could help determine what seems more plausible as an explanation (or it may be an unknown systematic effect). One should note that MLS profiles have been validated in the past for this region (see Boyd et al., 2007), and even though this comparison was versus microwave ground-based data as well, the agreement seemed to be significantly better than implied by the manuscript under review here.

Nevertheless, even if one can accept some systematic difference as large as 50%, why are the temporal tendencies sometimes different in Fig. 5? A more extended "reach" into the uppermost region from GROMOS (where MLS may follow a priori more) could explain the larger variability and larger values seen in Fig. 5 for GROMOS (especially at night). It seems more difficult to explain how one curve goes up for certain weeks while the other curve is coming down (or is flat), although I suspect that differences in sensitivity and resolution can account for much of this (What else could it be? This is not just an absolute value issue). For example, Fig. 3 in Sonneman et al. (2007) shows that different altitudes in a model simulation of ozone exhibit different temporal changes, so this could explain the results in the manuscript here, in theory, with different sensitivities to different altitudes (in turn, the variations can be caused by rapid wintertime changes in dynamics, temperature, and H2O, as mentioned in the above reference). In the manuscript, Fig. 4 does not include scatter plots or correlation coefficients for pressures less than 0.6 hPa, but the results are undoubtedly not as satisfying as at lower altitudes.

There are also a significant number of details to clean up (see below).

Smaller or more detailed comments:

Page 1, Line 10 (P1L1): the mean relative difference [singular] and its standard deviation increase with altitude up to 50% at 70 km. (I assume you mean that both the bias and the standard deviation are > 50%).

P1L15: not sure what is meant by "anomaly" here (better to use words like "wintertime enhancement").

P1L19/20: "...are its independence from solar irradiation and..."

P1L22/23: I suggest more concise wording, e.g. "Stratospheric ozone, in spite of its small abundance, plays a beneficial role by absorbing..."

P2L1, I would delete "Thus" at the beginning of the sentence.

P2L23: Suggested wording, "source of odd-hydrogen, coupled with no decrease [or no change] in the production of odd-oxygen..."

P2L29: a short discussion, and the conclusions are summarised in Section 5.

P3L26: Is the estimate of the a priori contribution not (more precisely) equal to 1 - the area, rather than the area itself? Then also, "We consider that the retrieval range is reliable where the true state dominates over the a priori information, ... I would note that this new retrieval characteristic is indeed quite different from past GROMOS papers, where it was not as well characterized near 0.05 hPa, but showing how the new and old retrieval compare, both in biases and in temporal behavior, would be very useful in order for the reader to decide how these are different (and how different versus MLS also). It is not immediately clear what helps to provide the extra information at high altitudes that was not present in earlier retrievals (clarify please). Usually this can come if one adds spectral channels, for example, or if one changes the smoothing characteristics in the retrievals (obtaining noisier retrievals but with more vertical information). In this respect, you quote the vertical resolution of the new retrieval, so comparing that to the old version would be useful as well.

P4L4: For the heading, why not capitalize "Microwave Limb Sounder" also?

Proper documentation/reference for the MLS data should be included. For example, the MLS website points to Data Quality Documentation (Livesey et al.) for version 4 data (including how to properly screen the data), and there are past references for validation as well (including Boyd et al., JGR, 2007, mentioned here already).

P5L2: Change "relies" to "lies".

P5L13: Change altitudes to altitude. Also, the last sentence in section 3 does not convey anything new and could be easily deleted.

P6L2: typo in "Germany".

P6L14: Change "shown" to "show"; delete "the" before "Figure 6".

P6L17: I suggest "although the latter data exhibit larger amplitudes".

P6L18: whereas at Lindau, winter-to-summer values vary by a factor of 2-3...

P6L19: definition of the MMM being restricted to high latitudes, we can report its observation with a smaller amplitude at mid-latitudes.

P6L23: Change "spaced-based" to "space-based".

P6L26: "we report good agreement between the new retrieval..."

P6L27: Change "Further" to "Furthermore".

Fig 2. I would say "The middle panel shows the mean relative difference..." Also, The mean absolute difference and its uncertainty (blu area) are displayed in the right panel. [with a period after the last word in the Fig. captions]. By the way, more needs to be clarifued here: is this for daytime or nighttime (presumably not) or for an average of day and night? The red line could be made thinner to allow one to see the blue line below it, or make the red line dashed maybe.

Fig. 3: Is this for nighttime data only or both averaged (it may not matter too much at these lower altitudes but still worth clarifying)?

Fig. 4: Same question as for Fig. 3 (same answer presumably).

Fig. 5: Change "ans the second panel" to "and the second panel".

---

## Referee Comment (RC2) · Anonymous Referee #2 · 2 May 2017

This paper has two main goals: to present a new version of the retrieval algorithm of the GROMOS dataset and to illustrate novel results concerning the diurnal variation of mesospheric O3 as observed by GROMOS. GROMOS provides an important long-term dataset for monitoring stratospheric ozone and therefore both points are important and make this work worth publishing in ACP or, possibly, AMT. However, I think this manuscript needs important improvements before publication.

For the first goal, my main objection is that the authors should show how the new retrievals compare with the previous ones. A comparison between the two versions of the dataset would provide to potential readers the necessary information on the improvements of the new dataset with respect to the GROMOS dataset presented in

the past (also very recently, for example by Moreira et al., 2016, in this same special issue). Additionally, such a comparison would offer a qualitative overview on how these new retrievals compare with other satellite-based or in situ datasets used in previous validation efforts of the GROMOS dataset.

For the second goal, I find that the main novelty contained in this manuscript is not in the observation of the MMM of O3 at mid-latitudes, but the fact that this specific long-term dataset (GROMOS data) displays it. It would therefore be interesting to show how well GROMOS does the job, i.e., how well GROMOS depicts the true state of the tertiay O3 which could be possibly represented by MLS original (high resolution) profiles.

Specific comments

Pg 1

Ln 13-15: This sentence presents a repetition that should be removed.

Pg 2

Ln 8-10: If GROMOS data have been validated in the past what is the need of an additional comparison with Aura/MLS? Differently, if the comparison with MLS serves as a validation of the new retrieval version, then a comparison of the new version with previous versions should also be present.

Ln 23-24: Awkward sentence

Ln 27: I would remove this sentence, or place it elsewhere.

Pg 3

Ln 9: What apriori information are you referring to? Temperature and pressure profiles? What about the ozone apriori profile?

Ln 18: Why do you have a systematic bias in the spectral measurements?

Ln 19: Even though the authors cite earlier papers describing in more details the technical aspects of the measurements, I think Figure 1 should still show an example of the spectrum measured and specify whether the 1-hour average spectrum is binned before deconvolving it. Are all channels binned in groups? Also those near the line center? This is critical for the high altitude comparison. Additionally, maybe a table similar to Table 1 of Moreira et al., 2015, would be a useful reminder of the main characteristics of GROMOS.

Ln 22: In figure 1, apriori and retrieved profiles are terribly close. I am aware that in the altitude region where the retrieval algorithm is the most sensitive the apriori has a very small impact on the profile retrieved, yet it would be nice to see it. Most readers don't know and will wonder what's the point of the measurement if the climatology from other datasets already provides you with the true state.

Pg 4

Ln 1: How is this an improvement with respect to the older version? Again, a comparison with the previous retrieval version is necessary.

Ln 19: Are these criteria consistent? The spatial requirement seems particularly generous compared to the temporal one. How far does a parcel of stratospheric air travel in one hour? A mesospheric one? Would a stricter spatial criterion improve your comparison results in the upper stratosphere/mesosphere? In other words, you should motivate your choices of coincident criteria.

Ln 21: I suggest "to" instead of "with the compliance of"

Pg 5

Line numbers indicated in the manuscript don't seem to be correct here and in the next page. I will refer to the actual line number.

Ln 1: I am not sure what this sentence implies. Are you suggesting that either the ground-based or the satellite-based data are inevitably faulty at high altitudes? Additionally, if I am not mistaken, the manuscripts you cite are either on SOMORA retrievals

(which reach 55 km at the most) or GROMOS itself. Are you suggesting that the present relatively large discrepancy in the GROMOS-MLS comparison at high altitude is likely to be due to GROMOS? If this is correct just say so.

Ln 4: I would write: "For an overview on the differences between coincident profiles, ..."

Ln 11: I would quantify the "almost perfect" with the slope of the linear fit. Second to last sentence in Section 3 : Could this be due to the spatial coincidence criterion? Last sentence in Section 3: I would suggest to postpone this last sentence to the conclusions section.

Pg 6

Line numbers indicated in the manuscript don't seem to be correct here or in the previous page. I will refer to the actual line number.

Ln 2: This needs to be better explained. Specifically, what part of your results agree with the work of Sonnemann 2007 and what doesn't. The fact that one dataset can peak at values that are twice as much as those of GROMOS seems an important difference. Do their data have a better vertical resolution? Retrievals that reach higher altitudes? Can you briefly address this difference?

Ln 4-8: I would remove these two sentences as they were already stated in the introduction

Ln 19: I would explicitly state what this anomaly is. Last two sentences in Section 4: It is not clear whether you ascribe the difference from Sonnemann et al. to the fact that Lindau is at higher latitudes. If this is the case, I would object that 5 degrees latitude cannot make this large difference in mesospheric ozone values and that a latitude of $51.7°$ N is not much higher than $47°$N.

Ln 27: Please, rephrase avoiding the repetition.

Ln 29: Together with the relative difference I would quote here also the absolute one, which is less than 0.2 ppmv, on average (if I read correctly from figure 2). Last sentence: I would specify what the anomaly is also here in the conclusions

Figure 1

- I would add a panel with the GROMOS 1-hour spectrum.

- I would enlarge, make it longer, the X-axis of the 3rd panel (maintaining the range 10-70 km).

Figure 2

- Would it be useful to show two separate averages, one for the daytime and one for the nighttime comparison?

- I would reduce the range of the X-axis of the middle plot to be from -60% to 60%

- I would use the same vertical unit (altitude or/and pressure) in all the figures or, even better, use both of them all the times. In figure 1 there's altitude, in figure 2 there's pressure.

Figure 3

- I would make these plots much larger, removing one or two pressure levels if necessary.

- Please specify in the caption the number of points involved in the moving average

Figure 4

- Same comment as for Figure 3: I would make these plots much larger, removing one or two pressure levels if necessary.

- I would add the numbers m and q in the equation y=mx+q for each linear fit, or at least the slope m.

- I am surprised by the relatively low correlation value at 0.617 hPa. By looking at figure 3 I was expecting a better result. Any comment?

Figure 5

- It would be useful to see a comparison of averaged nighttime vertical profiles, not just level 0.05 hPa, in order to establish, for example, whether the MLS O3 peak is at higher altitudes.

- As a matter of fact, it would be useful to see a comparison of GROMOS mesospheric profiles also with the averaged MLS original (not weighted with GROMOS AVK) night-time profiles, in order to understand the capabilities of GROMOS to spot the MMM with the "correct" intensity at the "correct" altitude.

- It would be best if line colors in the various figures were consistent, e.g., MLS always in red, GROMOS always in blue, and so on. In particular, maybe colors in Figure 5 could be changed (GROMOS in blue and cyan, MLS in red and orange?)

- Again, please in the caption state how many points are included in the average

- In the bottom plot I would add the standard deviation of the mean for both GROMOS and MLS.

Figure 6

- Given that the daytime mesospheric ozone at 0.05 hPa is relatively constant, the night to day ratio provides more or less the same information already present in Figure 5. Maybe I am wrong, but then the authors should make an effort in discussing this figure a little more.

---

## Referee Comment (RC3) · Anonymous Referee #3 · 23 May 2017

General comments.

This paper discusses comparisons of observations of mesospheric ozone between two established sensors, one ground-based, the other in low Earth orbit. Particular focus is given to the diurnal cycle seen around 70km in both datasets. The paper seems ideally suited to the special issue for NDACC. However, I'm less convinced that this paper belongs in ACP rather than AMT. Indeed, this manuscript feels like it sits exactly in the grey zone between them. If it were more of a "GROMOS v150 validation" paper it would clearly belong in AMT, but it is too lacking in detail to be that. If the focus was more on trying to understand why the two different sensors (plus the one at Lindau) report different behavior for the mesospheric diurnal cycle in ozone, then it
might be more clearly aimed at ACP (though, as many of those differences may well be instrumental in nature, such a discussion would still retain suitability for AMT). In either case, I feel important detail is lacking.

As it is, the paper presents differences between GROMOS and Aura MLS observations, but makes little attempt to explore their origins, nor even to comment on whether the magnitude of these differences is reasonable, given the differences in approach/performance between the two instruments. I feel greater effort needs to be made to explore these issues further for this paper to be a valuable addition to the field. For example, it's possible the discrepancy relates to differences in latitudinal sampling between MLS and GROMOS (see note on this issue below). A study of the latitudinal variability in the amplitude, based on MLS observations, could be used to quantify the degree to which the latitudinal sampling differences can account for the different amplitudes observed. I suggest the authors strengthen their analysis with some more consideration of such factors and an attempt to quantify (or at least bound) the potential contributors.

In addition, the paper is lacking in detail in several areas (some quite key) as discussed below.

The writing would benefit from some attention by a copy editor, as some of the choices of phraseology are awkward. I've pointed out some, but not all of these, and made suggestions for improvement in some places.

—————————————————————————————————-

More specific comments (of varying degrees of import).

(As noted by other reviewers, the line numbers in the manuscript are incorrect in some places. In contrast with the other reviewer, however, I'm going to continue to use them to index lines, for convenience. So in my [the authors] numbering scheme, the first lines on each page vary from 1 to -2.)

—- Page 1

Global note, I believe that it should be "Aura MLS" rather than "Aura/MLS"

line 4: "for the retrieval of" is odd wording: "A new version of the ozone profile retrievals..."

Line 8: Shouldn't it be "GROMOS and Aura MLS profiles agree within 3% on average for ..", or "Average GROMOS and ..." or "On average, GROMOS and ..."?

Lines 12/13: The sentence that spans these lines is poorly worded. "This behavior is related to..." is probably better. Also "On the other hand" is an inappropriate way in which to begin the sentence that follows.

Line 19: "its" -> "their"

Line 22: The assertion that this family of measurements have been indispensable would benefit from some citations that back that point up.

—- Page 2

Line 2: This sentence would also benefit from citations also (e.g., to some of the foundation documents for NDACC, or to GCOS [or similar] reports).

Line 10: "Furthermore" is inappropriate here. It's generally used when introducing a third or greater point, not for a second point. I suggest "In addition, we have ..." or "We have also,..."

Line 11: Badly constructed sentence. As written it sounds like there are two diurnal variations, one unspecified one, and one in mesospheric ozone, the amplitude of which you investigated.

Line 13/14. This explanation could be more complete, specifically, it would be good to give the timescale for the recombination. Presumably its ∼hours not ∼minutes, but needs to be made clear.

Line 14: "Moreover" feels like the wrong word here. "In addition..." might be better.

Line 18: "an effect occuring at" -> "a phenomenon that occurs at"

Line 22: comma needed between "and" and "since"

Lines 23/24: Badly worded sentence. Suggest: "The lack of odd-hydrogen needed for the catalytic depletion of odd-oxygen, in conjunction with an unchanged rate of odd oxygen production, leads to an increase in odd-oxygen"

Regarding the discussion in this section of the paper, the more conventional way to frame it is to list some relevant reactions and then talk about the processes that give rise to maxima and diurnal cycles etc. in terms of those reactions. So we'd have sentences along the lines of "Lack of sunlight inhibits generation of odd hydrogen via reaction X, leading to enhancement in odd oxygen abundances due to continued production by reaction Y", or something similar. The authors might want to consider taking that approach.

—- Page 3

Section 2.1. This section would benefit from having a few more details concerning the instrument. In particular, no information is given on the bandwidth of the observed spectrum, the spectral resolution, or the receiver noise temperature etc. These are all key parameters needed to get a sense of the measurement system. A plot showing a sample spectrum and associated error bars would be most welcome. For example, there's little point talking about adding 0.5K to the noise here or there without giving the reader a sense of how big the $T_{rec}/\sqrt{B\tau}$ number is. At what altitude does Doppler broadening start to dominate over pressure broadening for this line?

Also, presumably the retrievals need to assume a temperature (and height?) profile. Some information on where that is taken from, and the sensitivity of the result to it would be useful to give.

Line 8: Is the ozone a priori really taken from the ECMWF analysis? How useful is that

up to 70km, what is it based on. A reference would be good.

Line 13: You tell us that v150 has a constant a priori, but don't say how it behaved in 2021, it would be useful to know.

Line 13: "optimizing" in what sense, what were you trying to optimize? The vertical range, resolution, what? [Or should you change the "and" on the same line to "by"?]

Line 15: This discussion is a little confusing. Earlier parts of the paper give the impression that this study of the diurnal cycle was, at least partly, enabled by the new GROMOS data version. However, here you talk about the new version being focused on improvements in the lower stratosphere. If there were improvements in the mesosphere, it would be best to be more specific about what they are and which of the changes (presumably among those discussed above) brought those improvements about.

Lines 17/18: You need to define all of the terms in these equations, and give us the numbers for $T_{rec}$, B and tau.

Line 23: "The AVKs are multiplied by 4 in figure 1 in order to..."

Line 24: AVK -> AVKs

—- Page 4

Line 5 (your numbers): "our location" -> "Bern" or "the GROMOS measurement location" or similar.

Line 13: Suggest you make this a "displayed" equation rather than an "inline" one. Also, conventionally vectors are in lower case. If using LaTeX suggest _{\text{GROMOS}} (amsmath.sty) rather than _{GROMOS}, it give more suitable letter spacing (similarly for MLS).

Line 15: Surely Tsou is not the first such reference. Cite others, or at least put "e.g.," in front.

Line 19: More major point here. 8 degrees / 800km is a very large coincidence window, particularly given the ~165km along track spacing for MLS measurements. While you might need this on some days, when GROMOS falls in the gaps between the MLS orbits, on other days you'll get ~5 coincident observations. However, you do not tell us what you do in such circumstances. Do you compare your one GROMOS profile to all five? Do you pick the closest one? Do you average the five profiles together to give one comparison? What are the impacts of your choice on the subsequent analyses? More detail is needed here if readers are to be able to correctly interpret the results that follow.

Line 30: I'm a little bit wary of using the term absolute difference, more particularly in the caption for Figure 2, where you use the term "mean absolute difference". It could be taken to mean the mean of the unsigned difference, |a-b|. Perhaps simply say "mixing ratio difference"?

—- Page 5

Lines 2 and 3 (counting from -2): At face value, the 30-day smoothing and 4-day filtering appear to be contradictory. If the 30 data points are for 30 days worth of observations, then surely such a smoothing is going to filter far more aggressively than 4 days? Are there more than 30 points per day? Is this related to the issue of having more multiple MLS matches to a single GROMOS measurement? If so, this needs to be made much clearer. Plus, the impact of this smoothing is going to vary quite significantly depending on how many points there are on a given day. Why not simply smooth on a daily rather than a point-by-point basis (average of all differences within an n-day window)? Again, all this needs to be much more clearly described.

Line 8: "almost perfect" is very much in the eye of the beholder, and in my eye your scatter plots are far from it. To me "almost perfect" is at the >0.999 level of correlation, where the points are all but indistinguishable from the 1:1 line, with perhaps just one or two strays. I suggest you use more measured language.

Line 9: Odd way to phrase it, simply say that the black line is close to the green one to one line.

Line 21: "variation is also expected"

—- Page 6

Lines -2 to 2: As discussed above, more discussion is needed here. Some more investigation is needed as to why the amplitudes of the cycles are so different. You don't even tell us if we should be surprised by this level of disagreement. Note that the MLS averaging kernels imply not insignificant vertical smoothing at these altitudes for this instrument too. When taken in conjunction with the possible latitudinal gradient, are there plausible reasons to explain the differences based on sampling etc. alone, or is the only feasible explanation some instrumental/calibration difference? If nothing else, raise these questions and identify a route to answering them. Could the diurnal cycle in temperature (and thus the pressure/height relationship) play any role in this (from a measurement characteristics point of view rather than an atmospheric science one)? This manuscript would greatly benefit from an analysis, or at least an identification, of all the potential factors involved.

Lines 13-15: This discussion is unclear, at least to me. If the orange points are smoothed by 10 points, is that 10 days? How does this number related to the ~7 years between 2009 and 2016. I don't get how the 10-point and 30-point smoothings are related.

—- Figures

In general, all the figures use overly heavy line thicknesses. While it may be OK for the lines themselves (though rather on the heavy side), the linewidth used is far to heavy for the axes. Also the font should be slightly (~20-50%) larger, and perhaps not bold, for greater clarity.

Figure 2: Suggest "mean absolute difference" -> "mean mixing ratio difference". Also,

how is "its uncertainty" (last line) defined? Do you mean standard deviation?

---

## Author Comment (AC1) · 30 Jun 2017

**Response to anonymous referee #1**

Lorena Moreira

June 30, 2017

We would like to thank Referee #1 for the careful reading of our manuscript and for providing very constructive comments which certainly helped to improve the manuscript. This document includes all the referee's comments as well as our replies to every one of them. The changes in the manuscript are shown in blue and the text simply removed is crossed out in red.

As the **General Comments** from the Referee #1 are also mentioned in the **Smaller or more detailed comments** we will answer them separately in this section.

**Smaller or more detailed comments**

1. Comments from the referee: P1L10: The mean relative difference [singular] and its standard deviation increase with altitude up to 50% at 70 km. (I assume you mean that both the bias and the standard deviation are > 50%).

**Author's response:**

We have performed major changes in the comparison method. The criterion for spatial coincidence is now that horizontal distances between the sounding volumes of the satellite and the ground station have to be smaller than  $1^{\circ}$  in latitude and  $8^{\circ}$  in longitude. In addition, we have calculated the mean relative difference profile and the VMR difference profile separating daytime and nighttime values.

**Author's changes in the manuscript:**

P1L10: On average, GROMOS and MLS comparisons show agreement generally over 20% in the lower stratosphere and within 2% in the middle and upper stratosphere for both daytime and nighttime, whereas in the mesosphere the mean relative difference is below 40% at daytime and below 15% at nighttime.

P4L17: The selected criterion for spatial coincidence is that horizontal distances between the sounding volumes of the satellite and the ground station have to be smaller than 1° in latitude and 8° in longitude. The present study extends over the period from July 2009 to November 2016 and covers the stratosphere and middle mesosphere from 50 to 0.05 hPa (from 21 to 70 km), and according to the spatial and temporal criteria, more than 2800 coincident profiles are available for the comparison. Figure 3a and Figure 3b show the mean ozone profiles of the collocated and coincident measurements of GROMOS (blue line), MLS convolved (red line) and MLS original (green line) at daytime and nighttime, respectively. The relative difference profile in percent given by  $(\mathbf{x}_{\text{MLS,low}} - \mathbf{x}_{\text{GROMOS}})/\mathbf{x}_{\text{GROMOS}}$  is displayed

in the middle panel of both Figure 3a and Figure 3b along with the standard deviation of the differences (blue area). The green line delimits the  $\pm 10\%$  area. The mean profile of the VMR differences is shown in the right panel of both Figure 3. The mean relative differences and the VMR differences at daytime (nighttime) are over 20% or 0.5 ppm (15% or 0.4 ppm) in the lower stratosphere and decreasing with altitude up to 0.7% or 0.02 ppm (2% or 0.06 ppm) at the stratopause and increasing with altitude up to 38% or 0.085 ppm (15% or 0.12 ppm) at 0.05 hPa (70 km). We conclude from Figure 3 that during nighttime GROMOS measures more O3 VMR (ppm) than MLS except for the lower stratosphere, where MLS measures more O3 VMR (ppm) than GROMOS, both at daytime and nighttime. Nevertheless in the mesosphere GROMOS measures more O3 VMR (ppm) than MLS, both at daytime and nighttime.

Figure 3: Mean ozone profiles recorded by GROMOS (blue line), MLS convolved (red line) and MLS original (green line) for the time interval between July 2009 and November 2016 are shown in the left panels of both daytime and nighttime Figures. The blue area (GROMOS) and the red area (MLS) are the standard deviations of the coincident measurements. The middle panels show the mean relative difference profile between data of both instruments, GROMOS as reference. The blue areas in the middle panels represent the standard deviation of the differences. The green lines in the middle panel delimit the  $\pm$  10% area. The mean VMR difference profile and its standard deviation (blue area) are displayed in the right panels of both daytime and nighttime, Figure 3a and Figure 3b, respectively

P6L24: The agreement between measurements coincident in space and time for both data records is within 2% (0.06 ppm) between 30 and 50 km (15–0.7 hPa) increasing up to 20% (0.5 ppm) at 20 km (50 hPa), for both daytime and nighttime. In the mesosphere the difference increases up to 38% (0.085 ppm) at daytime and up to 15% (0.12 ppm) at nighttime at 70 km (0.05 hPa).

2. Comments from the referee: P1L15: not sure what is meant by "anomaly" here (better to use words like "wintertime enhancement").

**Author's response:**

We agree on the referee's comment and the text has been modified according to it.

We have decided to remove this line (P1L15).

**Author's changes in the manuscript:**

P1L15: On the other hand, the amplitude of the diurnal variation, night-to-day ratio (NDR), is not as strong as the observed one at higher latitudes, nevertheless we observe the winter anomaly of the night-to-day ratio.
P6L16: ... the expected wintertime enhancement of the NDR
P6L29: Moreover, the wintertime enhancement of nighttime ...

3. Comments from the referee: P1L19/20: "... are its independence from solar irradiation and ..."

**Author's response:**

No comments.

Author's changes in the manuscript: P1L19/20: ... are its independence from ...

- 4. Comments from the referee: P1L22/23: I suggest more concise wording, e.g. "Stratospheric ozone, in spite of its small abundance, plays a beneficial role by absorbing ..."
  - Author's response:

We agree on the referee's comment. The text has been modified according to it.

- Author's changes in the manuscript: P1L22/23: Stratospheric ozone, in spite of its small abundance, plays a beneficial role by absorbing ...
- 5. Comments from the referee: P2L1, I would delete "Thus" at the beginning of the sentence.

**Author's response:**

No comments.

Author's changes in the manuscript: P2L1: ... of the atmosphere. Continuous ...

6. **Comments from the referee:** P2L23: Suggested wording, "source of odd-hydrogen, coupled with no decrease [or no change] in the production of odd-oxygen..."

**Author's response:**

We agree on the referee's comment. The text has been modified according to it.

Author's changes in the manuscript: P2L20: Marsh et al. (2001) interpreted the tertiary peak by considering that in the middle mesosphere during winter, with solar zenith angle close to 90°, the atmosphere becomes optically thick to UV radiation at wavelengths below 185 nm and, since photolysis of water vapour (Reaction 1) is the primary source of odd-hydrogen, reduced UV radiation results in less odd-hydrogen. The lack of odd-hydrogen needed for the catalytic depletion of odd-oxygen (Reactions 2, 3 and 4), in conjunction with an unchanged rate of odd oxygen production (Reaction 5), leads to an increase in odd-oxygen. This results in higher ozone concentration because atomic oxygen recombination (Reaction 6) remains as a significant source of ozone in the mesosphere. Additionally, Hartogh et al. (2004) extended the interpretation by considering the very slow decrease of the ozone dissociation (Reaction 7) rate with increasing solar zenith angle.

$$H_2O + h\nu(\lambda

Figure 1: Mean ozone profiles retrieved by version 2021 (red line in the left panel) and by version 150 (blue line in the left panel) measured by GROMOS during the period from July 2009 to November 2016. The blue area (v150) and the red area (v2021) are the standard deviations of the ozone VMR. The mean relative difference profile (blue line) and the standard deviation of the differences (blue area) are represented in the middle panel, using the new version as reference. The green line delimits the  $\pm 10\%$  area. In the right panel is shown the VMR difference profile along with its standard deviation

**Author's changes in the manuscript:**

**P4L2: The Aura Microwave Limb Sounder**

P4L5: The satellite overpasses the GROMOS measurement location (at northern midlatitudes) twice a day, approximately around noon and midnight. The standard product for ozone is derived from MLS radiance measurements near 240 GHz. The vertical resolution of the ozone profiles ranges from 3 km in the stratosphere to 6 km in the mesosphere (Schwartz et al., 2008). The present study has used ozone profiles from version 4.2. A summary of the quality of version 4.2 Aura MLS Level 2 data can be found in Livesey et al. (2016). Details about the Aura mission can be found in Waters et al. (2006).

10. Comments from the referee: P5L2: Change "relies" to "lies".

**Author's response:**

We have removed this sentence.

- Author's changes in the manuscript: P5L1: This result is in agreement with other comparisons performed between ground-based microwave radiometers and spaced-based instruments above Switzerland, where the bias among data sets relied within 5–10 % in the stratosphere and up to 50% towards the mesosphere (Studer et al., 2013; Barras et al., 2009; Hocke et al., 2007; Dumitru et al., 2006; Calisesi et al., 2005).
- 11. Comments from the referee: P5L13: Change altitudes to altitude. Also, the last sentence in section 3 does not convey anything new and could be easily deleted.

**Author's response:**

No comments.

**Author's changes in the manuscript:**

P5L13: ... for the altitude above ...

P5L14: To sum up we can reiterate the fairly good agreement obtained for the comparison between ozone VMR profiles recorded by the ground-based instrument (GROMOS) and by the spaced-based instrument (Aura/MLS) during the time interval between July 2009 and November 2016 for the altitude range from 20 to 70 km.

12. Comments from the referee: P6L2: typo in 'Germany".

**Author's response:**

Thanks for spotting. We have corrected this.

Author's changes in the manuscript: P6L2: ... (Germany, ...

13. Comments from the referee: P6L14: Change "shown" to "show"; delete "the" before 'Figure 6".

Author's response:

No comments.

- Author's changes in the manuscript: P6L14: ... the second panel of Figure 7 show a moving average over 7 data points (1 week) with the aim to clarify the understanding of Figure 7
- 14. Comments from the referee: P6L17: I suggest "although the latter data exhibit larger amplitudes".

Author's response: No comments.

- Author's changes in the manuscript: P6L17: ..., although the latter data exhibit larger amplitudes.
- 15. Comments from the referee: P6L18: whereas at Lindau, winter-to-summer values vary by a factor of 2–3 ...

**Author's response:**

No comments.

- Author's changes in the manuscript: P6L18: ..., whereas at Lindau, winter-tosummer values vary by a factor of 2–3 at 70 km ...
- 16. Comments from the referee: P6L19: definition of the MMM being restricted to high latitudes, we can report its observation with a smaller amplitude at mid-latitudes.

**Author's response:**

No comments.

- Author's changes in the manuscript: P6L19: Thus, despite the definition of the MMM being restricted to high latitudes, we can report its observation with a smaller amplitude at mid-latitudes.
- 17. Comments from the referee: P6L23: Change "space-based" to "space-based".

Author's response:

Thanks for spotting. We have corrected this.

Author's changes in the manuscript: P6L23: ... by the space-based ...

18. Comments from the referee: P6L26: "we report good agreement between the new retrieval..."

Author's response:

No comments.

- Author's changes in the manuscript: P6L26: In general terms, we report good agreement between the new retrieval ...
- 19. Comments from the referee: P6L27: Change "Further" to "Furthermore".

**Author's response:**

No comments.

Author's changes in the manuscript: P6L27: Furthermore, we observe

20. Comments from the referee: Fig 2. I would say "The middle panel shows the mean relative difference..." Also, The mean absolute difference and its uncertainty (blu area) are displayed in the right panel. [with a period after the last word in the Fig. captions]. By the way, more needs to be clarified here: is this for daytime or nighttime (presumably not) or for an average of day and night? The red line could be made thinner to allow one to see the blue line below it, or make the red line dashed maybe.

**Author's response:**

A new Figure 3 is displayed in the first comment. In this new Figure 3 the comparison between GROMOS and MLS was performed by separating daytime (Figure 3a) and nighttime (Figure 3b) values.

**Author's changes in the manuscript: Figure 3**

21. Comments from the referee: Fig. 3: Is this for nighttime data only or both averaged (it may not matter too much at these lower altitudes but still worth clarifying)?

**Author's response:**

Former Figure 3 is now Figure 4, and in both the data represented is the average between daytime and nighttime data.

- Author's changes in the manuscript: P5L4: For an overview on the differences between coincident profiles, the average over daytime and nighttime values of the ozone VMR (ppm) time series of GROMOS (blue line) and MLS (red line) are displayed in Figure 4 for different pressure levels.
- 22. Comments from the referee: Fig. 4: Same question as for Fig. 3 (same answer presumably).

**Author's response:**

Former Figure 4 is now Figure 5 and in both the data represented is the average between daytime and nighttime data.the data represented is the average between daytime and nighttime data.

---

## Author Comment (AC2) · 30 Jun 2017

**Response to anonymous referee #2**

Lorena Moreira

June 30, 2017

We are very thankful to the anonymous Referee #2 for the evaluation of our manuscript and for the valuable comments that helped significantly to improve the quality of the paper. We have revised the manuscript by following each one of your suggestions. Below we try to answer each comment. The changes in the manuscript are shown in blue and the text simply removed is crossed out in red.

**Specific comments**

1. Comments from the referee: Pg. 1, Ln 13-15: This sentence presents a repetition that should be removed.

**Author's response:**

We agree on the referee's comment and the text has been modified according to it.

- Author's changes in the manuscript: Pg. 1, Ln 13-15: On the other hand, the amplitude of the diurnal variation, night-to-day ratio (NDR), is not as strong as the observed one at higher latitudes, nevertheless we observe the winter anomaly of the night-to-day ratio.
- 2. Comments from the referee: Pg. 2, Ln 8-10: If GROMOS data have been validated in the past what is the need of an additional comparison with Aura MLS? Differently, if the comparison with MLS serves as a validation of the new retrieval version, then a comparison of the new version with previous versions should also be present.

**Author's response:**

We agree with the referee and we have performed a comparison between version 2021 and version 150 of the retrieval of GROMOS.

Author's changes in the manuscript: Pg. 3, Ln 12: Recently, we have developed a new retrieval version (version 150) with the aim to optimise the averaging kernels. The differences with the former version (version 2021) are in the a priori covariance matrix, in the measurement error and in the integration time of the retrieval. In version 2021 the diagonal elements of the a priori covariance matrix are variable relative errors ranging from 35% at 100 hPa to 28% in the lower stratosphere and increasing with altitude from 35% in the upper stratosphere up to 70% in the mesosphere. Meanwhile, in version 150 the a priori covariance matrix has a constant value for the diagonal elements of 2 ppm. For both retrieval versions the

[revised manuscript text omitted]

discrepancy in the GROMOS-MLS comparison at high altitude is likely to be due to GROMOS? If this is correct just say so.

Author's response:

We agree on the referee's comment and we have removed the sentence.

- Author's changes in the manuscript: Pg. 5, Ln 1: This result is in agreement with other comparisons performed between ground-based microwave radiometers and spaced-based instruments above Switzerland, where the bias among data sets relied within 5–10 % in the stratosphere and up to 50% towards the mesosphere (Studer et al., 2013; Barras et al., 2009; Hocke et al., 2007; Dumitru et al., 2006; Calisesi et al., 2005).
- 13. Comments from the referee: Pg. 5, Ln 4: I would write: "For an overview on the differences between coincident profiles, ..."
  - Author's response:

No comments.

- Author's changes in the manuscript: Pg. 5, Ln 4: For an overview on the differences between coincident profiles, ...
- 14. Comments from the referee: Pg. 5, Ln 11: I would quantify the "almost perfect" with the slope of the linear fit. Second to last sentence in Section 3 : Could this be due to the spatial coincidence criterion? Last sentence in Section 3: I would suggest to postpone this last sentence to the conclusions section.

**Author's response:**

We agree on the referee's comment therefore we have changed line 11 and we have removed the last sentence of Section 3.

**Author's changes in the manuscript:**

Pg. 5, Ln1 1: The black lines, linear regression lines of the observations, are close to the green one to one lines,  $O_3(MLS)=O_3(GROMOS)$ .

Pg. 5, Ln 17–19: To sum up we can reiterate the fairly good agreement obtained for the comparison between ozone VMR profiles recorded by the ground-based instrument (GROMOS) and by the spaced-based instrument (Aura/MLS) during the time interval between July 2009 and November 2016 for the altitude range from 20 to 70 km.

15. Comments from the referee: Pg. 6, Ln 2: This needs to be better explained. Specifically, what part of your results agree with the work of Sonnemann 2007 and what doesn't. The fact that one dataset can peak at values that are twice as much as those of GROMOS seems an important difference. Do their data have a better vertical resolution? Retrievals that reach higher altitudes? Can you briefly address this difference?

**Author's response:**

Our results on the annual variation of mesospheric ozone at Bern are in agreement with the ones observed at Lindau by Sonnemann et al. (2007). The result disagrees in the amplitudes of the annual variation however according to Sonnemann et al. (2007), the MMM is an effect occurring at high latitudes close to the polar night terminator around 72 km altitude during nighttime in the winter half of the year and extends into middle latitudes with decreasing amplitude. Sonnemann et al. (2007) show nighttime ozone mixing ratio at Lindau up to 80 km. The upper altitude limit for the retrieval of ozone at 142 GHz measured by GROMOS is approximately 75 km, due to the fact that height-resolved information cannot be retrieved in the Doppler broadening domain since the line width does not depend on altitude. We set our altitude limit up to 70 km where the measurement response is  $\sim 1$ , therefore we do not have contribution from the a priori.

Author's changes in the manuscript: Pg. 6, Ln 1–2: Our results on the annual variation of mesospheric ozone at ...

Pg. 6, Ln 3: Disagreements appear in the amplitudes ...

16. Comments from the referee: Pg. 6, Ln 4-8: I would remove these two sentences as they were already stated in the introduction

**Author's response:**

No comments.

Author's changes in the manuscript: Pg. 6, Ln 4-8: This maximum of mesospheric ozone during nighttime in winter is related to the middle mesospheric maximum of ozone (MMM) (e.g., Sonnemann et al., 2007; Hartogh et al., 2004) also known as the tertiary ozone maximum (e.g., Sofieva et al., 2009; Degenstein et al., 2005; Marsh et al., 2001). During winter, the photodissociation rate of water is reduced at high latitudes which leads to a decrease of catalytic ozone depletion by odd hydrogen. 17. Comments from the referee: Pg. 6, Ln 19: I would explicitly state what this anomaly is. Last two sentences in Section 4: It is not clear whether you ascribe the difference from Sonnemann et al. to the fact that Lindau is at higher latitudes. If this is the case, I would object that 5° latitude cannot make this large difference in mesospheric ozone values and that a latitude of 51.7 °N is not much higher than 47°N.

**Author's response:**

We acknowledge that "winter anomaly" is maybe not the best appellation so we have changed for "wintertime enhancement".

According to Sonnemann et al. (2007), the MMM is an effect occurring at high latitudes close to the polar night terminator around 72 km altitude during nighttime in the winter half of the year and extends into middle latitudes with decreasing amplitude. The observed sharp decrease of the amplitude of the MMM of ozone is due to the strong latitudinal gradient between high and middle latitudes. In fact, it is surprising that we can observe the effect of MMM at our latitude. Therefore, the difference in latitude between Lindau and Bern may have such impact in the amplitudes of the annual variability of mesospheric ozone due to the MMM. However it could also be due to some other effects like for example, differences in the retrieval algorithms between Bern and Lindau, different instruments used to perform the measurements, different calculation methods...

Author's changes in the manuscript: Pg. 1, Ln 15: On the other hand, the amplitude of the diurnal variation, night-to-day ratio (NDR), is not as strong as the observed one at higher latitudes, nevertheless we observe the winter anomaly of the night-to-day ratio.

Pg. 6, Ln 19: ... the expected wintertime enhancement of the NDR

Pg. 6, Ln 32: Moreover, the wintertime enhancement of nighttime ...

Pg. 6, Ln 5: Nevertheless, our results are expected since this maximum of mesospheric ozone during nightime in winter is related to the middle mesospheric maximum of ozone (MMM) and according to Sonnemann et al. (2007) its effect extends into midlatitudes with decreasing amplitude.

**18. Comments from the referee: Pg. 6, Ln 27: Please, rephrase avoiding the repetition.**

**Author's response:**

No comments.

- Author's changes in the manuscript: Pg. 6, Ln 27: the diurnal variability and its amplitude, the night-to-day ratio (NDR).
- 19. Comments from the referee: Pg. 6, Ln 29: Together with the relative difference I would quote here also the absolute one, which is less than 0.2 ppmv, on average (if I read correctly from figure 2). Last sentence: I would specify what the anomaly is also here in the conclusions

**Author's response:**

No comments.

Author's changes in the manuscript: Pg. 6, Ln 29: The agreement between measurements coincident in space and time for both data records is within 2% (0.06 ppm) between 30 and 50 km (15–0.7 hPa) increasing up to 20% (0.5 ppm) at 20

km (50 hPa), for both daytime and nighttime. In the mesosphere the difference increases up to 38% (0.085 ppm) at daytime and up to 15% (0.12 ppm) at night-time at 70 km (0.05 hPa).

Pg. 6, Ln 32: Moreover, the wintertime enhancement of nighttime ...

**20. Comments from the referee: Figure 1:**

- I would add a panel with the GROMOS 1-hour spectrum.
- I would enlarge, make it longer, the X-axis of the 3rd panel (maintaining the range 10-70 km).

**Author's response:**

As we highlighted previously, we have not performed any instrumental change, therefore we can refer to Moreira et al. (2015) for these details.

With all due respect to the referee we do not understand the reason for enlarging the X-axis of the 3rd panel (maintaining the range 10-70 km).

**Author's changes in the manuscript: No changes.**

**21. Comments from the referee: Figure 2:**

- Would it be useful to show two separate averages, one for the daytime and one for the nighttime comparison?
- $\bullet~{\rm I}$  would reduce the range of the X-axis of the middle plot to be from -60% to 60%
- I would use the same vertical unit (altitude or/and pressure) in all the figures or, even better, use both of them all the times. In figure 1 there's altitude, in figure 2 there's pressure.

**Author's response:**

We have calculated the mean relative difference profile and the VMR difference profile separating daytime and nighttime values.

In Figure 2 (former Figure 1) we use altitude units in order to help in the interpretation of what it is shown.

**Author's changes in the manuscript: See the new Figure 3.**

**22. Comments from the referee: Figure 3:**

- I would make these plots much larger, removing one or two pressure levels if necessary.
- Please specify in the caption the number of points involved in the moving average Figure 4

**Author's response:**

With all due respect to the referee we think that the plots are larger enough to be properly interpreted.

Former Figure 3 is now Figure 4 and the number of points involved in the moving average is 7 points.

Author's changes in the manuscript: Caption of Figure 4: Time series of averaged daytime and nighttime  $O_3$  VMR measurements of GROMOS (blue line) and MLS (red line) for the period from July 2009 to November 2016 at different pressure levels. An averaging kernel smoothing has been applied to the series of the MLS

measurements coincident in time and space with the GROMOS measurements. Both time series are smoothed over 7 points or 1 week in time by a moving average

**23. Comments from the referee: Figure 4:**

- Same comment as for Figure 3: I would make these plots much larger, removing one or two pressure levels if necessary.
- I would add the numbers m and q in the equation y=mx+q for each linear fit, or at least the slope m.
- I am surprised by the relatively low correlation value at 0.617 hPa. By looking at figure 3 I was expecting a better result. Any comment?

**Author's response:**

With all due respect to the referee we think that the plots are large enough to be properly interpreted.

In accordance with the referee wishes we add the slope of every linear fit in the titles of plots which form Figure 5 (former Figure 4).

In our opinion this "low" correlation value can be expected from the time series at 0.617 hPa shown in Figure 5 (former Figure 4) since GROMOS measures more  $O_3$  VMR (ppm) for most of the summers under assessment.

**Author's changes in the manuscript: Figure 5**

---

## Author Comment (AC3) · 30 Jun 2017

**Response to anonymous referee #3**

Lorena Moreira

June 30, 2017

We are very grateful to Referee #3 for the useful and valuable comments which provided insights that helped significantly to improve the manuscript. All proposed objections and suggestions have been taken into account and discussed. Below we try to answer every comment. The changes in the manuscript are shown in blue and the text simply removed is crossed out in red.

**More specific comments**

- 1. Comments from the referee: Page 1, line 4: "for the retrieval of" is odd wording: "A new version of the ozone profile retrievals..."
  - Author's response:

No comments.

- Author's changes in the manuscript: Page 1, line 3–4: A new version of the ozone profile retrievals has been ...
- 2. Comments from the referee: Page 1, line 8: Shouldn't it be "GROMOS and Aura MLS profiles agree within 3% on average for ...", or "Average GROMOS and ..." or "On average, GROMOS and ..."?
  - Author's response:

No comments.

- Author's changes in the manuscript: Page 1, line 8: On average, GROMOS
- 3. Comments from the referee: Page 1, lines 12/13: The sentence that spans these lines is poorly worded. "This behavior is related to ..." is probably better. Also "On the other hand" is an inappropriate way in which to begin the sentence that follows.
  - Author's response:

We agree on the referee's comment. The text has been modified according to it.

Author's changes in the manuscript: Page 1, lines 12/13: This behavior is related to ...

Page 1, lines 13/15: On the other hand, the amplitude of the diurnal variation, night-to-day ratio (NDR), is not as strong as the observed one at higher latitudes, nevertheless we observe the winter anomaly of the night-to-day ratio.

4. Comments from the referee: Page 1, line 19: "its"  $\rightarrow$  "their"

**Author's response:**

Thanks for spotting. We have corrected this.

- Author's changes in the manuscript: Page 1, line 19: information about their distribution ...
- 5. Comments from the referee: Page 1, line 22: The assertion that this family of measurements have been indispensable would benefit from some citations that back that point up.

Author's response:

No comments.

- Author's changes in the manuscript: Page 1, line 22: Measurements of ozone performed by this technique have been indispensable in monitoring changes in the ozone layer and improving the comprehension of the processes that control ozone abundances (e.g. Steinbrecht et al. 2009).
- 6. Comments from the referee: Page 2, line 2: This sentence would also benefit from citations also (e.g., to some of the foundation documents for NDACC, or to GCOS [or similar] reports).

Author's response: No comments.

- Author's changes in the manuscript: Page 2, line 2: Continuous long-term monitoring of ozone is essential for the detection of long-term trends of the stratospheric ozone layer (e.g. WMO, 2014).
- 7. Comments from the referee: Page 2, line 10: "Furthermore" is inappropriate here. It's generally used when introducing a third or greater point, not for a second point. I suggest "In addition, we have ..." or "We have also,..."

**Author's response:**

No comments.

Author's changes in the manuscript: Page 2, line 10: We have also performed ...

8. Comments from the referee: Page 2, line 11: Badly constructed sentence. As written it sounds like there are two diurnal variations, one unspecified one, and one in mesospheric ozone, the amplitude of which you investigated.

**Author's response:**

No comments.

- Author's changes in the manuscript: Page 2, line 11: We have also performed an analysis of the diurnal variation and its amplitude (night-to-day ratio) of middle mesospheric ozone, at 0.05 hPa (70 km).
- 9. Comments from the referee: Page 2, line 13/14. This explanation could be more complete, specifically, it would be good to give the timescale for the recombination. Presumably it's ~ hours not ~ minutes, but needs to be made clear.

**Author's response:**

We have changed the sentence.

- Author's changes in the manuscript: Page 2, line 13/14: Daytime production of atomic oxygen by photolysis of ozone (Reaction 7) and photolysis of molecular oxygen (Reaction 5) results in nighttime ozone production by recombination of atomic and molecular oxygen (Reaction 6).
- 10. Comments from the referee: Page 2, line 14: "Moreover" feels like the wrong word here. "In addition..." might be better.
  - Author's response: No comments.
  - Author's changes in the manuscript: Page 2, line 14: In addition, we observe
- 11. Comments from the referee: Page 2, line 18: "an effect occuring at"  $\rightarrow$  "a phenomenon that occurs at"
  - Author's response: No comments.
  - Author's changes in the manuscript: Page 2, line 18: ... the MMM is a phenomenon that occurs at ...
- 12. Comments from the referee: Page 2, line 22: comma needed between "and" and "since"

**Author's response:**

No comments.

- Author's changes in the manuscript: Page 2, line 22: ... 185 nm and, since photolysis ...
- 13. Comments from the referee: Page 2, lines 23/24: Badly worded sentence. Suggest: "The lack of odd-hydrogen needed for the catalytic depletion of odd-oxygen, in conjunction with an unchanged rate of odd oxygen production, leads to an increase in odd-oxygen".

Regarding the discussion in this section of the paper, the more conventional way to frame it is to list some relevant reactions and then talk about the processes that give rise to maxima and diurnal cycles etc. in terms of those reactions. So we'd have sentences along the lines of "Lack of sunlight inhibits generation of odd hydrogen via reaction X, leading to enhancement in odd oxygen abundances due to continued production by reaction Y", or something similar. The authors might want to consider taking that approach.

**Author's response:**

We agree on the referee's comment. The text has been modified according to it.

Author's changes in the manuscript: Page 2, lines 20/24: Marsh et al. (2001) interpreted the tertiary peak by considering that in the middle mesosphere during winter, with solar zenith angle close to 90°, the atmosphere becomes optically thick to UV radiation at wavelengths below 185 nm and, since photolysis of water vapour (Reaction 1) is the primary source of odd-hydrogen, reduced UV radiation results in less odd-hydrogen. The lack of odd-hydrogen needed for the catalytic depletion of odd-oxygen (Reactions 2, 3 and 4), in conjunction with an unchanged rate of odd oxygen production (Reaction 5), leads to an increase in odd-oxygen. This results in higher ozone concentration because atomic oxygen recombination (Reaction 6) remains as a significant source of ozone in the mesosphere. Additionally, Hartogh et al. (2004) extended the interpretation by considering the very slow decrease of the ozone dissociation (Reaction 7) rate with increasing solar zenith angle.

$$H_2O + h\nu(\lambda

Figure 1: Mean ozone profiles retrieved by version 2021 (red line in the left panel) and by version 150 (blue line in the left panel) measured by GROMOS during the period from July 2009 to November 2016. The blue area (v150) and the red area (v2021) are the standard deviations of the ozone VMR. The mean relative difference profile (blue line) and the standard deviation of the differences (blue area) are represented in the middle panel, using the new version as reference. The green line delimits the  $\pm 10\%$  area. In the right panel is shown the VMR difference profile along with its standard deviation

Page 4, line 1: In version 2021, the vertical resolution lies generally within 10–15 km in the stratosphere and increases with altitude to 20–25 km in the lower mesosphere. Between 20 to 52 km (50 to 0.5 hPa) the measurement response is higher than 0.8. For more details on version 2021 we refer to Moreira et al. (2015). Comparing the measurement response and the vertical resolution obtained by version 2021 and by version 150 we can conclude an improvement in the results retrieved by version 150. We assume that the changes performed in the a priori covariance matrix, in the measurement noise and in the integration time result in the improvement of the retrieval product, mainly observed in the lowermost and in the uppermost limit of the retrieved ozone VMR profile.

17. Comments from the referee: Page 3, line 13: "optimizing" in what sense, what

were you trying to optimize? The vertical range, resolution, what? [Or should you change the "and" on the same line to "by"?]

**Author's response:**

No comments.

- Author's changes in the manuscript: Page 3, line 13: ..., thus optimizing the averaging kernels by improving ...
- 18. Comments from the referee: Page 3, line 15: This discussion is a little confusing. Earlier parts of the paper give the impression that this study of the diurnal cycle was, at least partly, enabled by the new GROMOS data version. However, here you talk about the new version being focused on improvements in the lower stratosphere. If there were improvements in the mesosphere, it would be best to be more specific about what they are and which of the changes (presumably among those discussed above) brought those improvements about.

**Author's response:**

No comments.

- Author's changes in the manuscript: Page 3, line 15: ... the measurement response in the lower stratosphere and in the mesosphere.
- 19. Comments from the referee: Page 3, lines 17/18: You need to define all of the terms in these equations, and give us the numbers for  $T_{rec}$ , B and tau.

**Author's response:**

We have changed the sentence.

- Author's changes in the manuscript: Page 3, lines 17/18: The error of the measured brightness temperature,  $\Delta T_b$ , is due to noise fluctuations in the spectrum and is of the order of a few Kelvins in the line center and 0.5 K in the line wings of the spectrum.
- 20. Comments from the referee: Page 3, line 23: "The AVKs are multiplied by 4 in figure 1 in order to ..."

**Author's response:**

Thanks for spotting. We have corrected this. Former Figure 1 is now Figure 2.

- Author's changes in the manuscript: Page 3, line 23: The AVKs are multiplied by 4 in Figure 2 in order ...
- 21. Comments from the referee: Page 3, line 24: AVK  $\rightarrow$  AVKs

Author's response:

No comments.

Author's changes in the manuscript: Page 3, line 24: AVKs ...

22. Comments from the referee: Page 4, line 5 (your numbers): "our location"  $\rightarrow$  "Bern" or "the GROMOS measurement location" or similar.

**Author's response:**

No comments.

Author's changes in the manuscript: Page 4, line 5: The satellite overpasses the GROMOS measurement location (at northern midlatitudes) twice a day

23. Comments from the referee: Page 4, Line 13: Suggest you make this a "displayed" equation rather than an "inline" one. Also, conventionally vectors are in lower case. If using LaTeX suggest GROMOS (amsmath.sty) rather than GROMOS, it give more suitable letter spacing (similarly for MLS).

**Author's response:**

No comments.

Author's changes in the manuscript: Page 4, Line 13: The smoothed profile of MLS adjusted to the vertical resolution of GROMOS is expressed as:

 $\mathbf{x}_{\text{MLS,low}} = \mathbf{x}_{\text{a,GROMOS}} + \mathbf{AVK}_{\text{GROMOS}} \cdot (\mathbf{x}_{\text{MLS,high}} - \mathbf{x}_{\text{a,GROMOS}})$ (10)

being  $AVK_{GROMOS}$  is the averaging kernel matrix of GROMOS,  $\mathbf{x}_{MLS,high}$  is the measured Aura/MLS profile and  $\mathbf{x}_{a,GROMOS}$  is the a priori profile ...

24. Comments from the referee: Page 4, Line 15: Surely Tsou is not the first such reference. Cite others, or at least put "e.g.," in front.

**Author's response:**

No comments.

Author's changes in the manuscript: Page 4, Line 15: by e.g. Tsou et al. (1995).

25. Comments from the referee: Page 4, line 19: More major point here. 8°/800 km is a very large coincidence window, particularly given the ~165km along track spacing for MLS measurements. While you might need this on some days, when GROMOS falls in the gaps between the MLS orbits, on other days you'll get ~5 coincident observations. However, you do not tell us what you do in such circumstances. Do you compare your one GROMOS profile to all five? Do you pick the closest one? Do you average the five profiles together to give one comparison? What are the impacts of your choice on the subsequent analyses? More detail is needed here if readers are to be able to correctly interpret the results that follow.

**Author's response:**

We have performed major changes in the comparison method. The criterion for spatial coincidence is now that horizontal distances between the sounding volumes of the satellite and the ground station have to be smaller than  $1^{\circ}$  in latitude and  $8^{\circ}$  in longitude. Then, I have one profile of MLS to compare to one profile of GROMOS every time the temporal and spatial criteria is fulfilled. We define as nighttime (daytime) value the average between the values recorded within 2 hours around midnight (noon).

- Author's changes in the manuscript: Page 4, line 17: The selected criterion for spatial coincidence is that horizontal distances between the sounding volumes of the satellite and the ground station have to be smaller than 1° in latitude and 8° in longitude.
- 26. Comments from the referee: Page 4, line 30: I'm a little bit wary of using the term absolute difference, more particularly in the caption for Figure 2, where you use the term "mean absolute difference". It could be taken to mean the mean of the unsigned difference, |a-b|. Perhaps simply say "mixing ratio difference"?

**Author's response:**

We agree on the referee's comment.

- Author's changes in the manuscript: We have changed mean absolute difference for mean VMR difference everywhere.
- 27. Comments from the referee: Page 5, lines 2 and 3 (counting from -2): At face value, the 30-day smoothing and 4-day filtering appear to be contradictory. If the 30 data points are for 30 days worth of observations, then surely such a smoothing is going to filter far more aggressively than 4 days? Are there more than 30 points per day? Is this related to the issue of having more multiple MLS matches to a single GROMOS measurement? If so, this needs to be made much clearer. Plus, the impact of this smoothing is going to vary quite significantly depending on how many points there are on a given day. Why not simply smooth on a daily rather than a point-by-point basis (average of all differences within an n-day window)? Again, all this needs to be much more clearly described.

**Author's response:**

As we have changed the spatial criteria of coincidence the number of coincident profiles has changed as well. Therefore, now we have performed moving average over 7 points which corresponds to around 1 week. Performing a daily smoothing in the time series will produce noisy and unclear Figures and hence difficulties to interpret them.

Author's changes in the manuscript: Page 5, lines 2 and 3: Short temporal fluctuations (periods

Figure 4: Time series of averaged daytime and nighttime  $O_3$  VMR measurements of GROMOS (blue line) and MLS (red line) for the period from July 2009 to November 2016 at different pressure levels. An averaging kernel smoothing has been applied to the series of the MLS measurements coincident in time and space with the GROMOS measurements. Both time series are smoothed over 7 points or 1 week in time by a moving average

28. Comments from the referee: Page 5, line 8: "almost perfect" is very much in the eye of the beholder, and in my eye your scatter plots are far from it. To me "almost perfect" is at the > 0.999 level of correlation, where the points are all but indistinguishable from the 1:1 line, with perhaps just one or two strays. I suggest you use more measured language.

**Author's response:**

No comment.

Author's changes in the manuscript: Page 5, line 8: An almost perfect agreement

29. Comments from the referee: Page 5, line 9: Odd way to phrase it, simply say that the black line is close to the green one to one line.

**Author's response:**

We agree with the referee.

- Author's changes in the manuscript: Page 5, line 8–9: The black lines, linear regression lines of the observations, are close to the green one to one lines,  $O_3(MLS)=O_3(GROMOS)$ .
- 30. Comments from the referee: Page 5, line 21: "variation is also expected"

**Author's response:**

Thanks for spotting. We have corrected this.

- Author's changes in the manuscript: Page 5, line 21: ... therefore an annual variation is also expected
- 31. Comments from the referee: Page 6, lines -2 to 2: As discussed above, more discussion is needed here. Some more investigation is needed as to why the amplitudes of the cycles are so different. You don't even tell us if we should be surprised by this level of disagreement. Note that the MLS averaging kernels imply not insignificant vertical smoothing at these altitudes for this instrument too. When taken in conjunction with the possible latitudinal gradient, are there plausible reasons to explain the differences based on sampling etc. alone, or is the only feasible explanation some instrumental/calibration difference? If nothing else, raise these questions and identify a route to answering them. Could the diurnal cycle in temperature (and thus the pressure/height relationship) play any role in this (from a measurement characteristics point of view rather than an atmospheric science one)? This manuscript would greatly benefit from an analysis, or at least an identification, of all the potential factors involved.

**Author's response:**

According to Sonnemann et al. (2007), the MMM is an effect occurring at high latitudes close to the polar night terminator around 72 km altitude during nighttime in the winter half of the year and extends into middle latitudes with decreasing amplitude. The observed sharp decrease of the amplitude of the MMM of ozone is due to the strong latitudinal gradient between high and middle latitudes. In fact, it is surprising that we can observe the effect of MMM at our latitude. Therefore, the difference in latitude between Lindau and Bern may have such impact in the amplitudes of the annual variability of mesospheric ozone due to the MMM. However it could also be due to some other effects like for example, differences in the retrieval algorithms between Bern and Lindau, different instruments used to perform the measurements, different calculation methods...

- Author's changes in the manuscript: Page 6, line 4: Nevertheless, our results are the expected since this maximum of mesospheric ozone during nighttime in winter is related to the middle mesospheric maximum of ozone (MMM) and according to Sonnemann et al. (2007) its effect extends into midlatitudes with decreasing amplitude.
- 32. Comments from the referee: Page 6, lines 13-15: This discussion is unclear, at least to me. If the orange points are smoothed by 10 points, is that 10 days? How does this number related to the ~7 years between 2009 and 2016. I don't get how the 10-point and 30-point smoothings are related.
  - Author's response:

We have repeated the comparison by changing the spatial criteria of coincidence and now the number of coincident profiles has changed. In the first panel of Figure 7 (former Figure 6) the moving average is over 30 points, roughly 1 month and in the second panel we used a moving average over 7 points which corresponds to around 1 week. The purpose of the smoothing is to help the interpretation of the results.

Author's changes in the manuscript: Page 5, line 30: All time series displayed in both panels of Figure 6 have been smoothed in time by a moving average over 15 data points ( $\sim 1$  week).

Page 6, lines 13-15: ...under assessment. Both time series were smoothed in time by a moving average over 30 points ( $\sim 1 \text{ month}$ ). ... the second panel of Figure 7 show a moving average over 7 data points (1 week) with the aim to clarify the understanding of Figure 7. The Aura/MLS and the GROMOS series depicted in Figure 5 and Figure 6 have been smoothed in time by a moving average over 30 data points.

33. Comments from the referee: Figures: In general, all the figures use overly heavy line thicknesses. While it may be OK for the lines themselves (though rather on the heavy side), the line width used is far to heavy for the axes. Also the font should be slightly (~20-50%) larger, and perhaps not bold, for greater clarity. Figure 2: Suggest "mean absolute difference" → "mean mixing ratio difference". Also, how is 'its uncertainty" (last line) defined? Do you mean standard deviation?

**Author's response:**

We have calculated the mean relative difference profile and the VMR difference profile separating daytime and nighttime values, accordingly Figure 3 (former Figure 2) has changed.

Author's changes in the manuscript: Figure 3

---

## Author Comment (AC4) · 30 Jun 2017

**Response to anonymous referee #1**

Lorena Moreira

June 30, 2017

We would like to thank Referee #1 for the careful reading of our manuscript and for providing very constructive comments which certainly helped to improve the manuscript. This document includes all the referee's comments as well as our replies to every one of them. The changes in the manuscript are shown in blue and the text simply removed is crossed out in red.

As the **General Comments** from the Referee #1 are also mentioned in the **Smaller or more detailed comments** we will answer them separately in this section.

**Smaller or more detailed comments**

1. **Comments from the referee:** P1L10: The mean relative difference [singular] and its standard deviation increase with altitude up to 50% at 70 km. (I assume you mean that both the bias and the standard deviation are > 50%).

   **Author's response:**
   We have performed major changes in the comparison method. The criterion for spatial coincidence is now that horizontal distances between the sounding volumes of the satellite and the ground station have to be smaller than 1° in latitude and 8° in longitude. In addition, we have calculated the mean relative difference profile and the VMR difference profile separating daytime and nighttime values.

   **Author's changes in the manuscript:**
   P1L10: On average, GROMOS and MLS comparisons show agreement generally over 20% in the lower stratosphere and within 2% in the middle and upper stratosphere for both daytime and nighttime, whereas in the mesosphere the mean relative difference is below 40% at daytime and below 15% at nighttime.

   P4L17: The selected criterion for spatial coincidence is that horizontal distances between the sounding volumes of the satellite and the ground station have to be smaller than 1° in latitude and 8° in longitude. The present study extends over the period from July 2009 to November 2016 and covers the stratosphere and middle mesosphere from 50 to 0.05 hPa (from 21 to 70 km), and according to the spatial and temporal criteria, more than 2800 coincident profiles are available for the comparison. Figure 3a and Figure 3b show the mean ozone profiles of the collocated and coincident measurements of GROMOS (blue line), MLS convolved (red line) and MLS original (green line) at daytime and nighttime, respectively. The relative difference profile in percent given by $(\mathbf{x}_{\mathrm{MLS,low}} - \mathbf{x}_{\mathrm{GROMOS}})/\mathbf{x}_{\mathrm{GROMOS}}$ is displayed

in the middle panel of both Figure 3a and Figure 3b along with the standard deviation of the differences (blue area). The green line delimits the ± 10% area. The mean profile of the VMR differences is shown in the right panel of both Figure 3. The mean relative differences and the VMR differences at daytime (nighttime) are over 20% or 0.5 ppm (15% or 0.4 ppm) in the lower stratosphere and decreasing with altitude up to 0.7% or 0.02 ppm (2% or 0.06 ppm) at the stratopause and increasing with altitude up to 38% or 0.085 ppm (15% or 0.12 ppm) at 0.05 hPa (70 km). We conclude from Figure 3 that during nighttime GROMOS measures more $O_3$ VMR (ppm) than MLS except for the lower stratosphere, where MLS measures more $O_3$ VMR (ppm) than GROMOS, both at daytime and nighttime. Nevertheless in the mesosphere GROMOS measures more $O_3$ VMR (ppm) than MLS, both at daytime and nighttime.

[Figure]

[Figure]

(a) Daytime            (b) Nighttime

Figure 3: Mean ozone profiles recorded by GROMOS (blue line), MLS convolved (red line) and MLS original (green line) for the time interval between July 2009 and November 2016 are shown in the left panels of both daytime and nighttime Figures. The blue area (GROMOS) and the red area (MLS) are the standard deviations of the coincident measurements. The middle panels show the mean relative difference profile between data of both instruments, GROMOS as reference. The blue areas in the middle panels represent the standard deviation of the differences. The green lines in the middle panel delimit the ± 10% area. The mean VMR difference profile and its standard deviation (blue area) are displayed in the right panels of both daytime and nighttime, Figure 3a and Figure 3b, respectively

P6L24: The agreement between measurements coincident in space and time for both data records is within 2% (0.06 ppm) between 30 and 50 km (15–0.7 hPa) increasing up to 20% (0.5 ppm) at 20 km (50 hPa), for both daytime and nighttime. In the mesosphere the difference increases up to 38% (0.085 ppm) at daytime and up to 15% (0.12 ppm) at nighttime at 70 km (0.05 hPa).

2. **Comments from the referee:** P1L15: not sure what is meant by "anomaly" here (better to use words like "wintertime enhancement").

**Author's response:**
We agree on the referee's comment and the text has been modified according to it.

We have decided to remove this line (P1L15).

**Author's changes in the manuscript:**
P1L15:
P6L16: ... the expected wintertime enhancement of the NDR
P6L29: Moreover, the wintertime enhancement of nighttime ...

3. **Comments from the referee:** P1L19/20: "... are its independence from solar irradiation and ..."

   **Author's response:**
   No comments.

   **Author's changes in the manuscript:** P1L19/20: ... are its independence from ...

4. **Comments from the referee:** P1L22/23: I suggest more concise wording, e.g. "Stratospheric ozone, in spite of its small abundance, plays a beneficial role by absorbing ..."

   **Author's response:**
   We agree on the referee's comment. The text has been modified according to it.

   **Author's changes in the manuscript:** P1L22/23: Stratospheric ozone, in spite of its small abundance, plays a beneficial role by absorbing ...

5. **Comments from the referee:** P2L1, I would delete "Thus" at the beginning of the sentence.

   **Author's response:**
   No comments.

   **Author's changes in the manuscript:** P2L1: ... of the atmosphere. Continuous ...

6. **Comments from the referee:** P2L23: Suggested wording, "source of odd-hydrogen, coupled with no decrease [or no change] in the production of odd-oxygen..."

   **Author's response:**
   We agree on the referee's comment. The text has been modified according to it.

   **Author's changes in the manuscript:** P2L20: Marsh et al. (2001) interpreted the tertiary peak by considering that in the middle mesosphere during winter, with solar zenith angle close to 90°, the atmosphere becomes optically thick to UV radiation at wavelengths below 185 nm and, since photolysis of water vapour (Reaction 1) is the primary source of odd-hydrogen, reduced UV radiation results in less odd-hydrogen. The lack of odd-hydrogen needed for the catalytic depletion of odd-oxygen (Reactions 2, 3 and 4), in conjunction with an unchanged rate of odd oxygen production (Reaction 5), leads to an increase in odd-oxygen. This results in higher ozone concentration because atomic oxygen recombination (Reaction 6) remains as a significant source of ozone in the mesosphere. Additionally, Hartogh et al. (2004) extended the interpretation by considering the very slow decrease of the ozone dissociation (Reaction 7) rate with increasing solar zenith angle.

$$\text{H}_2\text{O} + \text{h}\nu(\lambda < 185nm) \longrightarrow \text{OH} + \text{O} \qquad (1)$$

$$O + OH \longrightarrow O_2 + H \tag{2}$$

$$H + O_2 + M \longrightarrow HO_2 + M \tag{3}$$

$$O + HO_2 \longrightarrow O_2 + OH \tag{4}$$

$$O_2 + h\nu(\lambda < 242nm) \longrightarrow O + O \tag{5}$$

$$O + O_2 + M \longrightarrow O_3 + M \tag{6}$$

$$O_3 + h\nu \longrightarrow O_2 + O \tag{7}$$

7. **Comments from the referee:** P2L29: a short discussion, and the conclusions are summarised in Section 5.

   **Author's response:**
   No comments.

   **Author's changes in the manuscript:** P2L29: a short discussion, and the conclusions ...

8. **Comments from the referee:** P3L26: Is the estimate of the a priori contribution not (more precisely) equal to 1 - the area, rather than the area itself? Then also, "We consider that the retrieval range is reliable where the true state dominates over the a priori information, ... I would note that this new retrieval characteristic is indeed quite different from past GROMOS papers, where it was not as well characterised near 0.05 hPa, but showing how the new and old retrieval compare, both in biases and in temporal behaviour, would be very useful in order for the reader to decide how these are different (and how different versus MLS also). It is not immediately clear what helps to provide the extra information at high altitudes that was not present in earlier retrievals (clarify please). Usually this can come if one adds spectral channels, for example, or if one changes the smoothing characteristics in the retrievals (obtaining noisier retrievals but with more vertical information). In this respect, you quote the vertical resolution of the new retrieval, so comparing that to the old version would be useful as well.

   **Author's response:**
   We agree with the referee, an estimation of the a priori contribution is 1 minus the area of the averaging kernels.
   In accordance with the referee wishes we have performed a comparison between version 2021 and version 150 of the retrieval of GROMOS.

   **Author's changes in the manuscript:** P3L26: AVKs are a representation of the weighting of information content of the retrieval parameters therefore an estimate of the a priori contribution to the retrieval can be obtained by 1 minus the area of the AVK (measurement response).
   P3L12: Recently, we have developed a new retrieval version (version 150) with the aim to optimise the averaging kernels. The differences with the former version (version 2021) are in the a priori covariance matrix, in the measurement error and in the integration time of the retrieval.
   In version 2021 the diagonal elements of the a priori covariance matrix are variable relative errors ranging from 35% at 100 hPa to 28% in the lower stratosphere and increasing with altitude from 35% in the upper stratosphere up to 70% in

the mesosphere. Meanwhile, in version 150 the a priori covariance matrix has a constant value for the diagonal elements of 2 ppm. For both retrieval versions the off-diagonal elements of the a priori covariance matrix exponentially decrease with a correlation length of 3 km.

Regarding the measurement noise, in version 2021 it is a constant error of 0.8 K whereas in version 150 we used a variable error depending on the tropospheric transmission:

$$\Delta T_b' = 0.5 + \frac{\Delta T_b}{e^{-\tau}} \qquad (8)$$

the error of the measured brightness temperature, $\Delta T_b$, is given by the radiometer equation:

$$\Delta T_b = \frac{T_b + T_{rec}}{\sqrt{\Delta f \cdot t_{int}}} \qquad (9)$$

The radiometer equation gives the resolution of the radiation measured, which is determined by the bandwidth of the individual spectrometer channels ($\Delta f$), by the integration time ($t_{int}$) and by the total power measured by the spectrometer. A constant error of 0.5 K is considered as a systematic bias of the spectra, due to spectroscopic errors and the water vapour continuum. The error of the brightness temperature ($\Delta T_b$) is of the order of a few Kelvins in the line centre and 0.5 K in the line wings of the spectrum. Therefore the measurement noise ($\Delta T_b'$) depends on the bandwidth of the spectrum and on the tropospheric transmittance. This is a more realistic approach for the retrieval than considering a constant measurement noise, resulting in an improvement in the retrieved ozone VMR in the lower stratosphere. The sampling time for version 150 is 1 hour and in case of version 2021 is 30 minutes. Longer integration time improves the retrieved ozone VMR at upper altitudes.

P4L1: In version 2021, the vertical resolution lies generally within 10–15 km in the stratosphere and increases with altitude to 20–25 km in the lower mesosphere. Between 20 to 52 km (50 to 0.5 hPa) the measurement response is higher than 0.8. For more details on version 2021 we refer to Moreira et al. (2015). Comparing the measurement response and the vertical resolution obtained by version 2021 and by version 150 we can conclude an improvement in the results retrieved by version 150. We assume that the changes performed in the a priori covariance matrix, in the measurement noise and in the integration time result in the improvement of the retrieval product, mainly observed in the lowermost and in the uppermost limit of the retrieved ozone VMR profile.

9. **Comments from the referee:** P4L4: For the heading, why not capitalise 'Microwave Limb Sounder" also? Proper documentation/reference for the MLS data should be included. For example, the MLS website points to Data Quality Documentation (Livesey et al.) for version 4 data (including how to properly screen the data), and there are past references for validation as well (including Boyd et al., JGR, 2007, mentioned here already).

**Author's response:**
No comments.

[Figure]

Figure 1: Mean ozone profiles retrieved by version 2021 (red line in the left panel) and by version 150 (blue line in the left panel) measured by GROMOS during the period from July 2009 to November 2016. The blue area (v150) and the red area (v2021) are the standard deviations of the ozone VMR. The mean relative difference profile (blue line) and the standard deviation of the differences (blue area) are represented in the middle panel, using the new version as reference. The green line delimits the ±10% area. In the right panel is shown the VMR difference profile along with its standard deviation

**Author's changes in the manuscript:**
  P4L2: **The Aura Microwave Limb Sounder**
  P4L5: The satellite overpasses the GROMOS measurement location (at northern midlatitudes) twice a day, approximately around noon and midnight. The standard product for ozone is derived from MLS radiance measurements near 240 GHz. The vertical resolution of the ozone profiles ranges from 3 km in the stratosphere to 6 km in the mesosphere (Schwartz et al., 2008). The present study has used ozone profiles from version 4.2. A summary of the quality of version 4.2 Aura MLS Level 2 data can be found in Livesey et al. (2016). Details about the Aura mission can be found in Waters et al. (2006).

10. **Comments from the referee:** P5L2: Change "relies" to "lies".

    **Author's response:**
      We have removed this sentence.

    **Author's changes in the manuscript:** P5L1:

11. **Comments from the referee:** P5L13: Change altitudes to altitude. Also, the last sentence in section 3 does not convey anything new and could be easily deleted.

    **Author's response:**
      No comments.

**Author's changes in the manuscript:**
P5L13: ... for the altitude above ...
P5L14:

12. **Comments from the referee:** P6L2: typo in 'Germany".

    **Author's response:**
    Thanks for spotting. We have corrected this.

    **Author's changes in the manuscript:** P6L2: ... (Germany, ...

13. **Comments from the referee:** P6L14: Change "shown" to "show"; delete "the" before 'Figure 6".

    **Author's response:**
    No comments.

    **Author's changes in the manuscript:** P6L14: ... the second panel of Figure 7 show a moving average over 7 data points (1 week) with the aim to clarify the understanding of Figure 7

14. **Comments from the referee:** P6L17: I suggest "although the latter data exhibit larger amplitudes".

    **Author's response:**
    No comments.

    **Author's changes in the manuscript:** P6L17: ..., although the latter data exhibit larger amplitudes.

15. **Comments from the referee:** P6L18: whereas at Lindau, winter-to-summer values vary by a factor of 2–3 ...

    **Author's response:**
    No comments.

    **Author's changes in the manuscript:** P6L18: ..., whereas at Lindau, winter-to-summer values vary by a factor of 2–3 at 70 km ...

16. **Comments from the referee:** P6L19: definition of the MMM being restricted to high latitudes, we can report its observation with a smaller amplitude at midlatitudes.

    **Author's response:**
    No comments.

    **Author's changes in the manuscript:** P6L19: Thus, despite the definition of the MMM being restricted to high latitudes, we can report its observation with a smaller amplitude at mid-latitudes.

17. **Comments from the referee:** P6L23: Change "spaced-based" to "space-based".

**Author's response:**
Thanks for spotting. We have corrected this.

**Author's changes in the manuscript:** P6L23: ... by the space-based ...

18. **Comments from the referee:** P6L26: "we report good agreement between the new retrieval..."

    **Author's response:**
    No comments.

    **Author's changes in the manuscript:** P6L26: In general terms, we report good agreement between the new retrieval ...

19. **Comments from the referee:** P6L27: Change "Further" to "Furthermore".

    **Author's response:**
    No comments.

    **Author's changes in the manuscript:** P6L27: Furthermore, we observe

20. **Comments from the referee:** Fig 2. I would say "The middle panel shows the mean relative difference..." Also, The mean absolute difference and its uncertainty (blu area) are displayed in the right panel. [with a period after the last word in the Fig. captions]. By the way, more needs to be clarified here: is this for daytime or nighttime (presumably not) or for an average of day and night? The red line could be made thinner to allow one to see the blue line below it, or make the red line dashed maybe.

    **Author's response:**
    A new Figure 3 is displayed in the first comment. In this new Figure 3 the comparison between GROMOS and MLS was performed by separating daytime (Figure 3a) and nighttime (Figure 3b) values.

    **Author's changes in the manuscript:** Figure 3

21. **Comments from the referee:** Fig. 3: Is this for nighttime data only or both averaged (it may not matter too much at these lower altitudes but still worth clarifying)?

    **Author's response:**
    Former Figure 3 is now Figure 4, and in both the data represented is the average between daytime and nighttime data.

    **Author's changes in the manuscript:** P5L4: For an overview on the differences between coincident profiles, the average over daytime and nighttime values of the ozone VMR (ppm) time series of GROMOS (blue line) and MLS (red line) are displayed in Figure 4 for different pressure levels.

22. **Comments from the referee:** Fig. 4: Same question as for Fig. 3 (same answer presumably).

    **Author's response:**
    Former Figure 4 is now Figure 5 and in both the data represented is the average between daytime and nighttime data.the data represented is the average between daytime and nighttime data.

[Figure]

Figure 4: Time series of averaged daytime and nighttime $O_3$ VMR measurements of GROMOS (blue line) and MLS (red line) for the period from July 2009 to November 2016 at different pressure levels. An averaging kernel smoothing has been applied to the series of the MLS measurements coincident in time and space with the GROMOS measurements. Both time series are smoothed over 7 points or 1 week in time by a moving average

**Author's changes in the manuscript:** P5L10: In Figure 5 are shown the scatter plots of averaged daytime and nighttime $O_3$ VMR measurements of GROMOS and MLS at the same pressure levels as Figure 4.

[Figure]

Figure 5: Scatter plots of coincident $O_3$ VMR averaged over daytime and nighttime measurements of GROMOS and MLS for the period from July 2009 to November 2016 at different pressure levels. The black line is the linear fit of both time series. The green line indicates the case of identity, $O_3(MLS)=O_3(GROMOS)$. $r$ values are correlation coefficients of the MLS and GROMOS time series

23. **Comments from the referee:** Fig. 5: Change "ans the second panel" to "and the second panel".

**Author's response:**

Thank for spotting. We have corrected this. Former Figure 5 is now Figure 6.

**Author's changes in the manuscript:** Caption of Figure 6: ... and the second panel ...

**Response to anonymous referee #2**

Lorena Moreira

June 30, 2017

We are very thankful to the anonymous Referee #2 for the evaluation of our manuscript and for the valuable comments that helped significantly to improve the quality of the paper. We have revised the manuscript by following each one of your suggestions. Below we try to answer each comment. The changes in the manuscript are shown in blue and the text simply removed is crossed out in red.

**Specific comments**

1. **Comments from the referee:** Pg. 1, Ln 13-15: This sentence presents a repetition that should be removed.

   **Author's response:**
   We agree on the referee's comment and the text has been modified according to it.

   **Author's changes in the manuscript:** Pg. 1, Ln 13-15:

2. **Comments from the referee:** Pg. 2, Ln 8-10: If GROMOS data have been validated in the past what is the need of an additional comparison with Aura MLS? Differently, if the comparison with MLS serves as a validation of the new retrieval version, then a comparison of the new version with previous versions should also be present.

   **Author's response:**
   We agree with the referee and we have performed a comparison between version 2021 and version 150 of the retrieval of GROMOS.

   **Author's changes in the manuscript:** Pg. 3, Ln 12: Recently, we have developed a new retrieval version (version 150) with the aim to optimise the averaging kernels. The differences with the former version (version 2021) are in the a priori covariance matrix, in the measurement error and in the integration time of the retrieval.
   In version 2021 the diagonal elements of the a priori covariance matrix are variable relative errors ranging from 35% at 100 hPa to 28% in the lower stratosphere and increasing with altitude from 35% in the upper stratosphere up to 70% in the mesosphere. Meanwhile, in version 150 the a priori covariance matrix has a constant value for the diagonal elements of 2 ppm. For both retrieval versions the

off-diagonal elements of the a priori covariance matrix exponentially decrease with a correlation length of 3 km.

Regarding the measurement noise, in version 2021 it is a constant error of 0.8 K whereas in version 150 we used a variable error depending on the tropospheric transmission:

$$\Delta T_b' = 0.5 + \frac{\Delta T_b}{e^{-\tau}} \tag{1}$$

the error of the measured brightness temperature, $\Delta T_b$, is given by the radiometer equation:

$$\Delta T_b = \frac{T_b + T_{rec}}{\sqrt{\Delta f \cdot t_{int}}} \tag{2}$$

The radiometer equation gives the resolution of the radiation measured, which is determined by the bandwidth of the individual spectrometer channels ($\Delta f$), by the integration time ($t_{int}$) and by the total power measured by the spectrometer. A constant error of 0.5 K is considered as a systematic bias of the spectra, due to spectroscopic errors and the water vapour continuum. The error of the brightness temperature ($\Delta T_b$) is of the order of a few Kelvins in the line centre and 0.5 K in the line wings of the spectrum. Therefore the measurement noise ($\Delta T_b'$) depends on the bandwidth of the spectrum and on the tropospheric transmittance. This is a more realistic approach for the retrieval than considering a constant measurement noise, resulting in an improvement in the retrieved ozone VMR in the lower stratosphere. The sampling time for version 150 is 1 hour and in case of version 2021 is 30 minutes. Longer integration time improves the retrieved ozone VMR at upper altitudes.

[Figure]

Figure 1: Mean ozone profiles retrieved by version 2021 (red line in the left panel) and by version 150 (blue line in the left panel) measured by GROMOS during the period from July 2009 to November 2016. The blue area (v150) and the red area (v2021) are the standard deviations of the ozone VMR. The mean relative difference profile (blue line) and the standard deviation of the differences (blue area) are represented in the middle panel, using the new version as reference. The green line delimits the ±10% area. In the right panel is shown the VMR difference profile along with its standard deviation

Pg. 4, Ln 1: In version 2021, the vertical resolution lies generally within 10–15 km in the stratosphere and increases with altitude to 20–25 km in the lower mesosphere. Between 20 to 52 km (50 to 0.5 hPa) the measurement response is higher than 0.8. For more details on version 2021 we refer to Moreira et al. (2015). Comparing the measurement response and the vertical resolution obtained by version 2021 and by version 150 we can conclude an improvement in the results retrieved by version 150. We assume that the changes performed in the a priori covariance matrix, in the measurement noise and in the integration time result in the improvement of the retrieval product, mainly observed in the lowermost and in the uppermost limit of the retrieved ozone VMR profile.

3. **Comments from the referee:** Pg. 2, Ln 23-24: Awkward sentence

   **Author's response:**
   No comments.

   **Author's changes in the manuscript:** Pg. 2, Ln 20-26: Marsh et al. (2001) interpreted the tertiary peak by considering that in the middle mesosphere during winter, with solar zenith angle close to $90°$, the atmosphere becomes optically thick to UV radiation at wavelengths below 185 nm and, since photolysis of water vapour (Reaction 3) is the primary source of odd-hydrogen, reduced UV radiation results in less odd-hydrogen. The lack of odd-hydrogen needed for the catalytic depletion of odd-oxygen (Reactions 4, 5 and 6), in conjunction with an unchanged rate of odd oxygen production (Reaction 7), leads to an increase in odd-oxygen. This results in higher ozone concentration because atomic oxygen recombination (Reaction 8) remains as a significant source of ozone in the mesosphere. Additionally, Hartogh et al. (2004) extended the interpretation by considering the very slow decrease of the ozone dissociation (Reaction 9) rate with increasing solar zenith angle.

$$H_2O + h\nu(\lambda < 185nm) \longrightarrow OH + O \tag{3}$$

$$O + OH \longrightarrow O_2 + H \tag{4}$$

$$H + O_2 + M \longrightarrow HO_2 + M \tag{5}$$

$$O + HO_2 \longrightarrow O_2 + OH \tag{6}$$

$$O_2 + h\nu(\lambda < 242nm) \longrightarrow O + O \tag{7}$$

$$O + O_2 + M \longrightarrow O_3 + M \tag{8}$$

$$O_3 + h\nu \longrightarrow O_2 + O \tag{9}$$

4. **Comments from the referee:** Pg. 2, Ln 27: I would remove this sentence, or place it elsewhere.

   **Author's response:**
   No comments.

   **Author's changes in the manuscript:** Pg. 2, Ln 27:

5. **Comments from the referee:** Pg. 3, Ln 9: What a priori information are you referring to? Temperature and pressure profiles? What about the ozone a priori profile?

   **Author's response:**
   We agree on the referee's comment. The text has been modified according to it.

   **Author's changes in the manuscript:** Pg. 3, Ln 9: The a priori profile of $O_3$ VMR required for the retrieval is taken from a monthly varying climatology from ECMWF reanalysis until available (70 km) and extended above by an Aura MLS climatology (2004 to 2011). The line shape used in the retrieval is the representation of the Voigt line profile from Kuntz (1997). Spectroscopic parameters to calculate the ozone absorption coefficients were taken from the JPL catalogue (Pickett et al., 1998) and the HITRAN spectroscopic database (Rothman et al., 1998) The atmospheric temperature and pressure profiles are taken from the 6 hourly of the European Centre for Medium-Range Weather Forecast (ECMWF) operational analysis data and are extended above 80 km by monthly mean temperatures of the CIRA-86 Atmosphere Model (Fleming et al., 1990).

6. **Comments from the referee:** Pg. 3, Ln 18: Why do you have a systematic bias in the spectral measurements?

   **Author's response:**
   We do have systematic biases in the spectral measurements due to spectroscopic errors and the water vapour continuum.

   **Author's changes in the manuscript:** Pg. 3, Ln 18: In addition, a constant error of 0.5 K is considered as a systematic bias of the spectra, due to spectroscopic errors and the water vapour continuum.

7. **Comments from the referee:** Pg. 3, Ln 19: Even though the authors cite earlier papers describing in more details the technical aspects of the measurements, I think Figure 1 should still show an example of the spectrum measured and specify whether the 1-hour average spectrum is binned before deconvolving it. Are all channels binned in groups? Also those near the line center? This is critical for the high altitude comparison. Additionally, maybe a table similar to Table 1 of Moreira et al., 2015, would be a useful reminder of the main characteristics of GROMOS.

   **Author's response:**
   The referee is right to ask about an example of the spectrum measured and about a table of the GROMOS instrument specifications, yet we have not performed any instrumental change, therefore we can refer to Moreira et al. (2015) for these details.
   The fast Fourier transform spectrometer (FFTS) has around 32768 channels and after the binning in frequency the number of points in frequency are 54 with high frequency resolution in the line centre compare to the line wings.

   **Author's changes in the manuscript:** No changes.

8. **Comments from the referee:** Pg. 3, Ln 22: In figure 1, a priori and retrieved profiles are terribly close. I am aware that in the altitude region where the retrieval

algorithm is the most sensitive the a priori has a very small impact on the profile retrieved, yet it would be nice to see it. Most readers don't know and will wonder what's the point of the measurement if the climatology from other datasets already provides you with the true state.

**Author's response:**

In the middle panel of Figure 2 (former Figure 1) are shown the averaging kernels and the area of the averaging kernels, called measurement response. The averaging kernels are a key quantity for the characterisation of the retrieved profiles. It describes how the retrieval smoothes the true state and how sensitive it is to the a priori profile. The averaging kernel lines in the middle panel are shown in colour to help their interpretation, for instance, the green line shows the kernel line at 30 km and we can clearly see that the kernel actually peaks at 30 km. The measurement response is an indicator of the sensitive altitude range of the retrieved profile, it accounts for the amount of information from the true state of the retrieved profile at a given altitude. The measurement response (MR) is shown in red in the middle panel. It is considered a reliable altitude range of the retrieval when the true state dominates over the a priori information, i.e. where the measurement response is larger than 0.8 (an a priori contribution smaller than 20%). The measurement response shown in Figure 2 is around 1 from 18 to 70 km. Therefore, from this we can conclude the retrieved profile of GROMOS measurements is actually the true state of the atmosphere above Bern and not an a priori representation of the true state obtained from a climatology of other datasets.

**Author's changes in the manuscript:** No changes.

9. **Comments from the referee:** Pg. 4, Ln 1: How is this an improvement with respect to the older version? Again, a comparison with the previous retrieval version is necessary

**Author's response:**

As previously mentioned we have performed a comparison between version 2021 and version 150 of the retrieval of GROMOS. See comment 2 for details on the changes in the manuscript.

**Author's changes in the manuscript:** See **Comment from the referee 2** for details on the changes in the manuscript.

10. **Comments from the referee:** Pg. 4, Ln 19: Are these criteria consistent? The spatial requirement seems particularly generous compared to the temporal one. How far does a parcel of stratospheric air travel in one hour? A mesospheric one? Would a stricter spatial criterion improve your comparison results in the upper stratosphere/mesosphere? In other words, you should motivate your choices of coincident criteria.

**Author's response:**

In accordance with the referee wishes we have performed major changes in the comparison method. The criterion for spatial coincidence is now that horizontal distances between the sounding volumes of the satellite and the ground station have to be smaller than 1° in latitude and 8° in longitude. In addition, we have calculated the mean relative difference profile and the VMR difference profile separating daytime and nighttime values.

**Author's changes in the manuscript:** Pg. 1, Ln 10: On average, GROMOS and MLS comparisons show agreement generally over 20% in the lower stratosphere and within 2% in the middle and upper stratosphere for both daytime and nighttime, whereas in the mesosphere the mean relative difference is below 40% at daytime and below 15% at nighttime.

Pg. 4, Ln 17: The selected criterion for spatial coincidence is that horizontal distances between the sounding volumes of the satellite and the ground station have to be smaller than 1° in latitude and 8° in longitude. The present study extends over the period from July 2009 to November 2016 and covers the stratosphere and middle mesosphere from 50 to 0.05 hPa (from 21 to 70 km), and according to the spatial and temporal criteria, more than 2800 coincident profiles are available for the comparison. Figure 3a and Figure 3b show the mean ozone profiles of the collocated and coincident measurements of GROMOS (blue line), MLS convolved (red line) and MLS original (green line) at daytime and nighttime, respectively. The relative difference profile in percent given by $(\mathbf{x}_{\mathrm{MLS,low}} - \mathbf{x}_{\mathrm{GROMOS}})/\mathbf{x}_{\mathrm{GROMOS}}$ is displayed in the middle panel of both Figure 3a and Figure 3b along with the standard deviation of the differences (blue area). The green line delimits the ± 10% area. The mean profile of the VMR differences is shown in the right panel of both Figure 3. The mean relative differences and the VMR differences at daytime (nighttime) are over 20% or 0.5 ppm (15% or 0.4 ppm) in the lower stratosphere and decreasing with altitude up to 0.7% or 0.02 ppm (2% or 0.06 ppm) at the stratopause and increasing with altitude up to 38% or 0.085 ppm (15% or 0.12 ppm) at 0.05 hPa (70 km). We conclude from Figure 3 that during nighttime GROMOS measures more $O_3$ VMR (ppm) than MLS except for the lower stratosphere, where MLS measures more $O_3$ VMR (ppm) than GROMOS, both at daytime and nighttime. Nevertheless in the mesosphere GROMOS measures more $O_3$ VMR (ppm) than MLS, both at daytime and nighttime.

Pg. 6, Ln 24: The agreement between measurements coincident in space and time for both data records is within 2% (0.06 ppm) between 30 and 50 km (15–0.7 hPa) increasing up to 20% (0.5 ppm) at 20 km (50 hPa), for both daytime and nighttime. In the mesosphere the difference increases up to 38% (0.085 ppm) at daytime and up to 15% (0.12 ppm) at nighttime at 70 km (0.05 hPa).

11. **Comments from the referee:** Pg. 4, Ln 21: I suggest "to" instead of "with the compliance of"

**Author's response:**
No comments.

**Author's changes in the manuscript:** Pg. 4, Ln 21: ... and according to the spatial and ...

12. **Comments from the referee:** Pg. 5, Ln 1: I am not sure what this sentence implies. Are you suggesting that either the ground-based or the satellite-based data are inevitably faulty at high altitudes? Additionally, if I am not mistaken, the manuscripts you cite are either on SOMORA retrievals (which reach 55 km at the most) or GROMOS itself. Are you suggesting that the present relatively large

[Figure]

[Figure]

(a) Daytime          (b) Nighttime

Figure 3: Mean ozone profiles recorded by GROMOS (blue line), MLS convolved (red line) and MLS original (green line) for the time interval between July 2009 and November 2016 are shown in the left panels of both daytime and nighttime Figures. The blue area (GROMOS) and the red area (MLS) are the standard deviations of the coincident measurements. The middle panels show the mean relative difference profile between data of both instruments, GROMOS as reference. The blue areas in the middle panels represent the standard deviation of the differences. The green lines in the middle panel delimit the ± 10% area. The mean VMR difference profile and its standard deviation (blue area) are displayed in the right panels of both daytime and nighttime, Figure 3a and Figure 3b, respectively

discrepancy in the GROMOS-MLS comparison at high altitude is likely to be due to GROMOS? If this is correct just say so.

**Author's response:**
We agree on the referee's comment and we have removed the sentence.

**Author's changes in the manuscript:** Pg. 5, Ln 1:

13. **Comments from the referee:** Pg. 5, Ln 4: I would write: "For an overview on the differences between coincident profiles, ..."

**Author's response:**
No comments.

**Author's changes in the manuscript:** Pg. 5, Ln 4: For an overview on the differences between coincident profiles, ...

14. **Comments from the referee:** Pg. 5, Ln 11: I would quantify the "almost perfect" with the slope of the linear fit. Second to last sentence in Section 3 : Could this be due to the spatial coincidence criterion? Last sentence in Section 3: I would suggest to postpone this last sentence to the conclusions section.

**Author's response:**

We agree on the referee's comment therefore we have changed line 11 and we have removed the last sentence of Section 3.

**Author's changes in the manuscript:**

Pg. 5, Ln1 1: The black lines, linear regression lines of the observations, are close to the green one to one lines, $O_3$(MLS)=$O_3$(GROMOS).

Pg. 5, Ln 17–19:

15. **Comments from the referee:** Pg. 6, Ln 2: This needs to be better explained. Specifically, what part of your results agree with the work of Sonnemann 2007 and what doesn't. The fact that one dataset can peak at values that are twice as much as those of GROMOS seems an important difference. Do their data have a better vertical resolution? Retrievals that reach higher altitudes? Can you briefly address this difference?

**Author's response:**

Our results on the annual variation of mesospheric ozone at Bern are in agreement with the ones observed at Lindau by Sonnemann et al. (2007). The result disagrees in the amplitudes of the annual variation however according to Sonnemann et al. (2007), *the MMM is an effect occurring at high latitudes close to the polar night terminator around 72 km altitude during nighttime in the winter half of the year and extends into middle latitudes with decreasing amplitude.* Sonnemann et al. (2007) show nighttime ozone mixing ratio at Lindau up to 80 km. The upper altitude limit for the retrieval of ozone at 142 GHz measured by GROMOS is approximately 75 km, due to the fact that height-resolved information cannot be retrieved in the Doppler broadening domain since the line width does not depend on altitude. We set our altitude limit up to 70 km where the measurement response is $\sim 1$, therefore we do not have contribution from the a priori.

**Author's changes in the manuscript:** Pg. 6, Ln 1–2: Our results on the annual variation of mesospheric ozone at ...

Pg. 6, Ln 3: Disagreements appear in the amplitudes ...

16. **Comments from the referee:** Pg. 6, Ln 4-8: I would remove these two sentences as they were already stated in the introduction

**Author's response:**

No comments.

**Author's changes in the manuscript:** Pg. 6, Ln 4-8: ~~This maximum of mesospheric ozone during nighttime in winter is related to the middle mesospheric maximum of ozone (MMM) (e.g., Sonnemann et al., 2007; Hartogh et al., 2004) also known as the tertiary ozone maximum (e.g., Sofieva et al., 2009; Degenstein et al., 2005; Marsh et al., 2001). During winter, the photodissociation rate of water is reduced at high latitudes which leads to a decrease of catalytic ozone depletion by odd hydrogen.~~

17. **Comments from the referee:** Pg. 6, Ln 19: I would explicitly state what this anomaly is. Last two sentences in Section 4: It is not clear whether you ascribe the difference from Sonnemann et al. to the fact that Lindau is at higher latitudes. If this is the case, I would object that 5° latitude cannot make this large difference in mesospheric ozone values and that a latitude of 51.7 °N is not much higher than 47°N.

    **Author's response:**
    We acknowledge that "winter anomaly" is maybe not the best appellation so we have changed for "wintertime enhancement".
    According to Sonnemann et al. (2007), *the MMM is an effect occurring at high latitudes close to the polar night terminator around 72 km altitude during nighttime in the winter half of the year and extends into middle latitudes with decreasing amplitude.* The observed sharp decrease of the amplitude of the MMM of ozone is due to the strong latitudinal gradient between high and middle latitudes. In fact, it is surprising that we can observe the effect of MMM at our latitude. Therefore, the difference in latitude between Lindau and Bern may have such impact in the amplitudes of the annual variability of mesospheric ozone due to the MMM. However it could also be due to some other effects like for example, differences in the retrieval algorithms between Bern and Lindau, different instruments used to perform the measurements, different calculation methods...

    **Author's changes in the manuscript:** Pg. 1, Ln 15:
    Pg. 6, Ln 19: ... the expected wintertime enhancement of the NDR
    Pg. 6, Ln 32: Moreover, the wintertime enhancement of nighttime ...
    Pg. 6, Ln 5: Nevertheless, our results are expected since this maximum of mesospheric ozone during nighttime in winter is related to the middle mesospheric maximum of ozone (MMM) and according to Sonnemann et al. (2007) its effect extends into midlatitudes with decreasing amplitude.

18. **Comments from the referee:** Pg. 6, Ln 27: Please, rephrase avoiding the repetition.

    **Author's response:**
    No comments.

    **Author's changes in the manuscript:** Pg. 6, Ln 27: the diurnal variability and its amplitude, the night-to-day ratio (NDR).

19. **Comments from the referee:** Pg. 6, Ln 29: Together with the relative difference I would quote here also the absolute one, which is less than 0.2 ppmv, on average (if I read correctly from figure 2). Last sentence: I would specify what the anomaly is also here in the conclusions

    **Author's response:**
    No comments.

    **Author's changes in the manuscript:** Pg. 6, Ln 29: The agreement between measurements coincident in space and time for both data records is within 2% (0.06 ppm) between 30 and 50 km (15–0.7 hPa) increasing up to 20% (0.5 ppm) at 20

km (50 hPa), for both daytime and nighttime. In the mesosphere the difference increases up to 38% (0.085 ppm) at daytime and up to 15% (0.12 ppm) at nighttime at 70 km (0.05 hPa).

Pg. 6, Ln 32: Moreover, the wintertime enhancement of nighttime ...

20. **Comments from the referee:** Figure 1:

- I would add a panel with the GROMOS 1-hour spectrum.
- I would enlarge, make it longer, the X-axis of the 3rd panel (maintaining the range 10-70 km).

**Author's response:**

As we highlighted previously, we have not performed any instrumental change, therefore we can refer to Moreira et al. (2015) for these details.

With all due respect to the referee we do not understand the reason for enlarging the X-axis of the 3rd panel (maintaining the range 10-70 km).

**Author's changes in the manuscript:** No changes.

21. **Comments from the referee:** Figure 2:

- Would it be useful to show two separate averages, one for the daytime and one for the nighttime comparison?
- I would reduce the range of the X-axis of the middle plot to be from -60% to 60%
- I would use the same vertical unit (altitude or/and pressure) in all the figures or, even better, use both of them all the times. In figure 1 there's altitude, in figure 2 there's pressure.

**Author's response:**

We have calculated the mean relative difference profile and the VMR difference profile separating daytime and nighttime values.

In Figure 2 (former Figure 1) we use altitude units in order to help in the interpretation of what it is shown.

**Author's changes in the manuscript:** See the new Figure 3.

22. **Comments from the referee:** Figure 3:

- I would make these plots much larger, removing one or two pressure levels if necessary.
- Please specify in the caption the number of points involved in the moving average Figure 4

**Author's response:**

With all due respect to the referee we think that the plots are larger enough to be properly interpreted.

Former Figure 3 is now Figure 4 and the number of points involved in the moving average is 7 points.

**Author's changes in the manuscript:** Caption of Figure 4: Time series of averaged daytime and nighttime $O_3$ VMR measurements of GROMOS (blue line) and MLS (red line) for the period from July 2009 to November 2016 at different pressure levels. An averaging kernel smoothing has been applied to the series of the MLS

23. **Comments from the referee:** Figure 4:

  - Same comment as for Figure 3: I would make these plots much larger, removing one or two pressure levels if necessary.
  - I would add the numbers m and q in the equation y=mx+q for each linear fit, or at least the slope m.
  - I am surprised by the relatively low correlation value at 0.617 hPa. By looking at figure 3 I was expecting a better result. Any comment?

**Author's response:**
With all due respect to the referee we think that the plots are large enough to be properly interpreted.

In accordance with the referee wishes we add the slope of every linear fit in the titles of plots which form Figure 5 (former Figure 4).

In our opinion this "low" correlation value can be expected from the time series at 0.617 hPa shown in Figure 5 (former Figure 4) since GROMOS measures more $O_3$ VMR (ppm) for most of the summers under assessment.

**Author's changes in the manuscript:** Figure 5

[Figure]

Figure 5: Scatter plots of coincident $O_3$ VMR measurements of GROMOS and MLS for the period from July 2009 to November 2016 at different pressure levels. The black line is the linear fit of both time series, and $m$ the slope of the linear fit. The green line indicates the case of identity, $O_3(MLS)=O_3(GROMOS)$. $r$ values are correlation coefficients of the MLS and GROMOS time series

24. **Comments from the referee:** Figure 5:

  - It would be useful to see a comparison of averaged nighttime vertical profiles, not just level 0.05 hPa, in order to establish, for example, whether the MLS $O_3$ peak is at higher altitudes.
  - As a matter of fact, it would be useful to see a comparison of GROMOS mesospheric profiles also with the averaged MLS original (not weighted with

GROMOS AVK) nighttime profiles, in order to understand the capabilities of GROMOS to spot the MMM with the "correct" intensity at the "correct" altitude.

- It would be best if line colors in the various figures were consistent, e.g., MLS always in red, GROMOS always in blue, and so on. In particular, maybe colors in Figure 5 could be changed (GROMOS in blue and cyan, MLS in red and orange?)
- Again, please in the caption state how many points are included in the average
- In the bottom plot I would add the standard deviation of the mean for both GROMOS and MLS.

**Author's response:**

We have analysed the MMM at different altitudes, and for instance, at 0.1 hPa ($\sim$ 63 km) the results are pretty similar to the ones obtained at 0.05 hPa, although with smaller amplitudes.

In accordance with the referee wishes we have repeated the Figure. Former Figure 5 is now Figure 6.

With all due respect to the referee we think that our colours choice for this Figure 6 is rather intuitive.

Regarding, the addition of the standard deviation of the mean for both data records we think that this choice would make the Figure noisy. The standard deviation of the mean is $\sim$0.3 ppm for GROMOS, $\sim$0.2 ppm for MLS convolved and $\sim$0.5 ppm for MLS original.

**Author's changes in the manuscript:** Pg. 5, Ln 27: The first panel of Figure 6 displays the $O_3$ VMR measured at noon (GROMOS in red, MLS convolved in orange and MLS original in magenta) and at midnight (GROMOS in blue, MLS convolved in cyan and MLS original in black) at 0.05 hPa (70 km) for the already mentioned time period. The original MLS data, i.e. not weighted with GROMOS AVKs, is shown in order to provide an insight of the observability of the effect of MMM at northern midlatitudes by GROMOS.

25. **Comments from the referee:** Figure 6: Given that the daytime mesospheric ozone at 0.05 hPa is relatively constant, the night to day ratio provides more or less the same information already present in Figure 5. Maybe I am wrong, but then the authors should make an effort in discussing this figure a little more.

**Author's response:**

We acknowledge that the night-to-day ratio (NDR) just provides information about the amplitude of the diurnal and seasonal variability of mesospheric ozone, nevertheless we want to keep it in the manuscript in order to be comparable with the study of Sonnemann et al. (2007).

**Author's changes in the manuscript:** No changes.

[Figure]

Figure 6: The first panel shows the diurnal variation of $O_3$ VMR measured at noon (GROMOS in red, MLS convolved in orange and MLS original in magenta) and at midnight (GROMOS in blue, MLS convolved in cyan and MLS original in black) at 0.05 hPa (70 km) and the second panel shows its evolution throughout the year averaged for the time interval under assessment (July 2009–November 2016). All time series are smoothed in time by a moving average over 15 points (1 week)

**Response to anonymous referee #3**

Lorena Moreira

June 30, 2017

We are very grateful to Referee #3 for the useful and valuable comments which provided insights that helped significantly to improve the manuscript. All proposed objections and suggestions have been taken into account and discussed. Below we try to answer every comment. The changes in the manuscript are shown in blue and the text simply removed is crossed out in red.

**More specific comments**

1. **Comments from the referee:** Page 1, line 4:"for the retrieval of" is odd wording: "A new version of the ozone profile retrievals..."

   **Author's response:**
   No comments.

   **Author's changes in the manuscript:** Page 1, line 3–4: A new version of the ozone profile retrievals has been ...

2. **Comments from the referee:** Page 1, line 8: Shouldn't it be "GROMOS and Aura MLS profiles agree within 3% on average for ...", or "Average GROMOS and ..." or "On average, GROMOS and ..."?

   **Author's response:**
   No comments.

   **Author's changes in the manuscript:** Page 1, line 8: On average, GROMOS

3. **Comments from the referee:** Page 1, lines 12/13: The sentence that spans these lines is poorly worded."This behavior is related to ..." is probably better. Also "On the other hand" is an inappropriate way in which to begin the sentence that follows.

   **Author's response:**
   We agree on the referee's comment. The text has been modified according to it.

   **Author's changes in the manuscript:** Page 1, lines 12/13: This behavior is related to ...
   Page 1, lines 13/15:

4. **Comments from the referee:** Page 1, line 19: "its" → "their"

**Author's response:**
Thanks for spotting. We have corrected this.

**Author's changes in the manuscript:** Page 1, line 19: information about their distribution ...

5. **Comments from the referee:** Page 1, line 22: The assertion that this family of measurements have been indispensable would benefit from some citations that back that point up.

**Author's response:**
No comments.

**Author's changes in the manuscript:** Page 1, line 22: Measurements of ozone performed by this technique have been indispensable in monitoring changes in the ozone layer and improving the comprehension of the processes that control ozone abundances (e.g. Steinbrecht et al. 2009).

6. **Comments from the referee:** Page 2, line 2: This sentence would also benefit from citations also (e.g., to some of the foundation documents for NDACC, or to GCOS [or similar] reports).

**Author's response:**
No comments.

**Author's changes in the manuscript:** Page 2, line 2: Continuous long-term monitoring of ozone is essential for the detection of long-term trends of the stratospheric ozone layer (e.g. WMO, 2014).

7. **Comments from the referee:** Page 2, line 10: "Furthermore" is inappropriate here. It's generally used when introducing a third or greater point, not for a second point. I suggest "In addition, we have ..." or "We have also,..."

**Author's response:**
No comments.

**Author's changes in the manuscript:** Page 2, line 10: We have also performed ...

8. **Comments from the referee:** Page 2, line 11: Badly constructed sentence. As written it sounds like there are two diurnal variations, one unspecified one, and one in mesospheric ozone, the amplitude of which you investigated.

**Author's response:**
No comments.

**Author's changes in the manuscript:** Page 2, line 11: We have also performed an analysis of the diurnal variation and its amplitude (night-to-day ratio) of middle mesospheric ozone, at 0.05 hPa (70 km).

9. **Comments from the referee:** Page 2, line 13/14. This explanation could be more complete, specifically, it would be good to give the timescale for the recombination. Presumably it's $\sim$ hours not $\sim$ minutes, but needs to be made clear.

**Author's response:**
We have changed the sentence.

**Author's changes in the manuscript:** Page 2, line 13/14: Daytime production of atomic oxygen by photolysis of ozone (Reaction 7) and photolysis of molecular oxygen (Reaction 5) results in nighttime ozone production by recombination of atomic and molecular oxygen (Reaction 6).

10. **Comments from the referee:** Page 2, line 14: "Moreover" feels like the wrong word here. "In addition..." might be better.

    **Author's response:**
    No comments.

    **Author's changes in the manuscript:** Page 2, line 14: In addition, we observe

11. **Comments from the referee:** Page 2, line 18: "an effect occuring at" → "a phenomenon that occurs at"

    **Author's response:**
    No comments.

    **Author's changes in the manuscript:** Page 2, line 18: ... the MMM is a phenomenon that occurs at ...

12. **Comments from the referee:** Page 2, line 22: comma needed between "and" and "since"

    **Author's response:**
    No comments.

    **Author's changes in the manuscript:** Page 2, line 22: ... 185 nm and, since photolysis ...

13. **Comments from the referee:** Page 2, lines 23/24: Badly worded sentence. Suggest: "The lack of odd-hydrogen needed for the catalytic depletion of odd-oxygen, in conjunction with an unchanged rate of odd oxygen production, leads to an increase in odd-oxygen".
    Regarding the discussion in this section of the paper, the more conventional way to frame it is to list some relevant reactions and then talk about the processes that give rise to maxima and diurnal cycles etc. in terms of those reactions. So we'd have sentences along the lines of "Lack of sunlight inhibits generation of odd hydrogen via reaction X, leading to enhancement in odd oxygen abundances due to continued production by reaction Y", or something similar. The authors might want to consider taking that approach.

    **Author's response:**
    We agree on the referee's comment. The text has been modified according to it.

    **Author's changes in the manuscript:** Page 2, lines 20/24: Marsh et al. (2001) interpreted the tertiary peak by considering that in the middle mesosphere during winter, with solar zenith angle close to 90°, the atmosphere becomes optically thick to UV radiation at wavelengths below 185 nm and, since photolysis of water vapour (Reaction 1) is the primary source of odd-hydrogen, reduced UV radiation results in less odd-hydrogen. The lack of odd-hydrogen needed for the catalytic depletion of odd-oxygen (Reactions 2, 3 and 4), in conjunction with an unchanged rate of odd oxygen production (Reaction 5), leads to an increase in odd-oxygen. This

results in higher ozone concentration because atomic oxygen recombination (Reaction 6) remains as a significant source of ozone in the mesosphere. Additionally, Hartogh et al. (2004) extended the interpretation by considering the very slow decrease of the ozone dissociation (Reaction 7) rate with increasing solar zenith angle.

$$\text{H}_2\text{O} + \text{h}\nu(\lambda < 185nm) \longrightarrow \text{OH} + \text{O} \tag{1}$$

$$\text{O} + \text{OH} \longrightarrow \text{O}_2 + \text{H} \tag{2}$$

$$\text{H} + \text{O}_2 + \text{M} \longrightarrow \text{HO}_2 + \text{M} \tag{3}$$

$$\text{O} + \text{HO}_2 \longrightarrow \text{O}_2 + \text{OH} \tag{4}$$

$$\text{O}_2 + \text{h}\nu(\lambda < 242nm) \longrightarrow \text{O} + \text{O} \tag{5}$$

$$\text{O} + \text{O}_2 + \text{M} \longrightarrow \text{O}_3 + \text{M} \tag{6}$$

$$\text{O}_3 + \text{h}\nu \longrightarrow \text{O}_2 + \text{O} \tag{7}$$

14. **Comments from the referee:** Page 3, Section 2.1. This section would benefit from having a few more details concerning the instrument. In particular, no information is given on the bandwidth of the observed spectrum, the spectral resolution, or the receiver noise temperature etc. These are all key parameters needed to get a sense of the measurement system. A plot showing a sample spectrum and associated error bars would be most welcome. For example, there's little point talking about adding 0.5K to the noise here or there without giving the reader a sense of how big the $\text{T}_{rec}$/sqrt(B tau) number is. At what altitude does Doppler broadening start to dominate over pressure broadening for this line? Also, presumably the retrievals need to assume a temperature (and height?) profile. Some information on where that is taken from, and the sensitivity of the result to it would be useful to give.

**Author's response:**
The referee is right to ask about more details concerning the instrument, yet for these details we refer to Moreira et al. (2015).
This 0.5 K added to the noise is due to spectroscopic errors and the water vapour continuum.
The Doppler broadening starts to dominate above 75 km, in case of ozone at 142 GHz.
We agree on the referee's comment about more information on the temperature and pressure profiles needed for the retrieval. The text has been modified according to it.

**Author's changes in the manuscript:** Page 3, line 18: In addition, a constant error of 0.5 K is considered as a systematic bias of the spectra, due to spectroscopic errors and the water vapour continuum.
Page 3, line 9: The a priori profile required for the retrieval is taken from a monthly varying climatology from ECMWF reanalysis until available (70 km) and extended above by an Aura MLS climatology (2004 to 2011). The line shape used in the retrieval is the representation of the Voigt line profile from Kuntz (1997). Spectroscopic parameters to calculate the ozone absorption coefficients were taken from the JPL catalogue (Pickett et al., 1998) and the HITRAN spectroscopic database (Rothman et al., 1998) The atmospheric temperature and pressure

profiles are taken from the 6 hourly of the European Centre for Medium-Range Weather Forecast (ECMWF) operational analysis data and are extended above 80 km by monthly mean temperatures of the CIRA-86 Atmosphere Model (Fleming et al., 1990).

15. **Comments from the referee:** Page 3, line 8: Is the ozone a priori really taken from the ECMWF analysis? How useful is that up to 70km, what is it based on. A reference would be good.

    **Author's response:**
    Yes, it is. The a priori ozone profile does not play a role since the measurement response, area of the averaging kernels, is equal to unity for the altitude range from 18 to 70 km.

    **Author's changes in the manuscript:** No changes.

16. **Comments from the referee:** Page 3, line 13: You tell us that v150 has a constant a priori, but don't say how it behaved in 2021, it would be useful to know.

    **Author's response:**
    In version 2021, as diagonal elements of the a priori covariance matrix we assume a relative error around 35% at 100 hPa. The error decreases in the lower stratosphere up to 28%. Then it increases linearly from 35% in the upper stratosphere to 70% in the lower mesosphere. The off-diagonal elements exponentially decrease with a correlation length of 3 km.
    We have performed a comparison between version 2021 and version 150 of the retrieval of GROMOS.

    **Author's changes in the manuscript:**
    Page 3, line 12: Recently, we have developed a new retrieval version (version 150) with the aim to optimise the averaging kernels. The differences with the former version (version 2021) are in the a priori covariance matrix, in the measurement error and in the integration time of the retrieval.
    In version 2021 the diagonal elements of the a priori covariance matrix are variable relative errors ranging from 35% at 100 hPa to 28% in the lower stratosphere and increasing with altitude from 35% in the upper stratosphere up to 70% in the mesosphere. Meanwhile, in version 150 the a priori covariance matrix has a constant value for the diagonal elements of 2 ppm. For both retrieval versions the off-diagonal elements of the a priori covariance matrix exponentially decrease with a correlation length of 3 km.
    Regarding the measurement noise, in version 2021 it is a constant error of 0.8 K whereas in version 150 we used a variable error depending on the tropospheric transmission:

$$\Delta T_b' = 0.5 + \frac{\Delta T_b}{e^{-\tau}} \qquad (8)$$

    the error of the measured brightness temperature, $\Delta T_b$, is given by the radiometer equation:

$$\Delta T_b = \frac{T_b + T_{rec}}{\sqrt{\Delta f \cdot t_{int}}} \qquad (9)$$

    The radiometer equation gives the resolution of the radiation measured, which is determined by the bandwidth of the individual spectrometer channels ($\Delta f$), by

the integration time ($t_{int}$) and by the total power measured by the spectrometer. A constant error of 0.5 K is considered as a systematic bias of the spectra, due to spectroscopic errors and the water vapour continuum. The error of the brightness temperature ($\Delta T_b$) is of the order of a few Kelvins in the line centre and 0.5 K in the line wings of the spectrum. Therefore the measurement noise ($\Delta T_b'$) depends on the bandwidth of the spectrum and on the tropospheric transmittance. This is a more realistic approach for the retrieval than considering a constant measurement noise, resulting in an improvement in the retrieved ozone VMR in the lower stratosphere. The sampling time for version 150 is 1 hour and in case of version 2021 is 30 minutes. Longer integration time improves the retrieved ozone VMR at upper altitudes.

[Figure]

Figure 1: Mean ozone profiles retrieved by version 2021 (red line in the left panel) and by version 150 (blue line in the left panel) measured by GROMOS during the period from July 2009 to November 2016. The blue area (v150) and the red area (v2021) are the standard deviations of the ozone VMR. The mean relative difference profile (blue line) and the standard deviation of the differences (blue area) are represented in the middle panel, using the new version as reference. The green line delimits the ±10% area. In the right panel is shown the VMR difference profile along with its standard deviation

Page 4, line 1: In version 2021, the vertical resolution lies generally within 10–15 km in the stratosphere and increases with altitude to 20–25 km in the lower mesosphere. Between 20 to 52 km (50 to 0.5 hPa) the measurement response is higher than 0.8. For more details on version 2021 we refer to Moreira et al. (2015). Comparing the measurement response and the vertical resolution obtained by version 2021 and by version 150 we can conclude an improvement in the results retrieved by version 150. We assume that the changes performed in the a priori covariance matrix, in the measurement noise and in the integration time result in the improvement of the retrieval product, mainly observed in the lowermost and in the uppermost limit of the retrieved ozone VMR profile.

17. **Comments from the referee:** Page 3, line 13: "optimizing" in what sense, what

were you trying to optimize? The vertical range, resolution, what? [Or should you change the "and" on the same line to "by"?]

**Author's response:**
No comments.

**Author's changes in the manuscript:** Page 3, line 13: ..., thus optimizing the averaging kernels by improving ...

18. **Comments from the referee:** Page 3, line 15: This discussion is a little confusing. Earlier parts of the paper give the impression that this study of the diurnal cycle was, at least partly, enabled by the new GROMOS data version. However, here you talk about the new version being focused on improvements in the lower stratosphere. If there were improvements in the mesosphere, it would be best to be more specific about what they are and which of the changes (presumably among those discussed above) brought those improvements about.

**Author's response:**
No comments.

**Author's changes in the manuscript:** Page 3, line 15: ... the measurement response in the lower stratosphere and in the mesosphere.

19. **Comments from the referee:** Page 3, lines 17/18: You need to define all of the terms in these equations, and give us the numbers for $T_{rec}$, B and tau.

**Author's response:**
We have changed the sentence.

**Author's changes in the manuscript:** Page 3, lines 17/18: The error of the measured brightness temperature, $\Delta T_b$, is due to noise fluctuations in the spectrum and is of the order of a few Kelvins in the line center and 0.5 K in the line wings of the spectrum.

20. **Comments from the referee:** Page 3, line 23: "The AVKs are multiplied by 4 in figure 1 in order to ..."

**Author's response:**
Thanks for spotting. We have corrected this. Former Figure 1 is now Figure 2.

**Author's changes in the manuscript:** Page 3, line 23: The AVKs are multiplied by 4 in Figure 2 in order ...

21. **Comments from the referee:** Page 3, line 24: AVK → AVKs

**Author's response:**
No comments.

**Author's changes in the manuscript:** Page 3, line 24: AVKs ...

22. **Comments from the referee:** Page 4, line 5 (your numbers): "our location" → "Bern" or "the GROMOS measurement location" or similar.

**Author's response:**
No comments.

**Author's changes in the manuscript:** Page 4, line 5: The satellite overpasses the GROMOS measurement location (at northern midlatitudes) twice a day

23. **Comments from the referee:** Page 4, Line 13: Suggest you make this a "displayed" equation rather than an "inline" one. Also, conventionally vectors are in lower case. If using LaTeX suggest $_{\mathrm{GROMOS}}$ (amsmath.sty) rather than $_{GROMOS}$, it give more suitable letter spacing (similarly for MLS).

**Author's response:**
No comments.

**Author's changes in the manuscript:** Page 4, Line 13: The smoothed profile of MLS adjusted to the vertical resolution of GROMOS is expressed as:

$$\mathbf{x}_{\mathrm{MLS,low}} = \mathbf{x}_{\mathrm{a,GROMOS}} + \mathbf{AVK}_{\mathrm{GROMOS}} \cdot (\mathbf{x}_{\mathrm{MLS,high}} - \mathbf{x}_{\mathrm{a,GROMOS}}) \qquad (10)$$

being $\mathbf{AVK}_{\mathrm{GROMOS}}$ is the averaging kernel matrix of GROMOS, $\mathbf{x}_{\mathrm{MLS,high}}$ is the measured Aura/MLS profile and $\mathbf{x}_{\mathrm{a,GROMOS}}$ is the a priori profile ...

24. **Comments from the referee:** Page 4, Line 15: Surely Tsou is not the first such reference. Cite others, or at least put "e.g.," in front.

**Author's response:**
No comments.

**Author's changes in the manuscript:** Page 4, Line 15: by e.g. Tsou et al. (1995).

25. **Comments from the referee:** Page 4, line 19: More major point here. 8°/800 km is a very large coincidence window, particularly given the ∼165km along track spacing for MLS measurements. While you might need this on some days, when GROMOS falls in the gaps between the MLS orbits, on other days you'll get ∼5 coincident observations. However, you do not tell us what you do in such circumstances. Do you compare your one GROMOS profile to all five? Do you pick the closest one? Do you average the five profiles together to give one comparison? What are the impacts of your choice on the subsequent analyses? More detail is needed here if readers are to be able to correctly interpret the results that follow.

**Author's response:**
We have performed major changes in the comparison method. The criterion for spatial coincidence is now that horizontal distances between the sounding volumes of the satellite and the ground station have to be smaller than 1° in latitude and 8° in longitude. Then, I have one profile of MLS to compare to one profile of GROMOS every time the temporal and spatial criteria is fulfilled. We define as nighttime (daytime) value the average between the values recorded within 2 hours around midnight (noon).

**Author's changes in the manuscript:** Page 4, line 17: The selected criterion for spatial coincidence is that horizontal distances between the sounding volumes of the satellite and the ground station have to be smaller than 1° in latitude and 8° in longitude.

26. **Comments from the referee:** Page 4, line 30: I'm a little bit wary of using the term absolute difference, more particularly in the caption for Figure 2, where you use the term "mean absolute difference". It could be taken to mean the mean of the unsigned difference, |a-b|. Perhaps simply say "mixing ratio difference"?

**Author's response:**
We agree on the referee's comment.

**Author's changes in the manuscript:** We have changed mean absolute difference for mean VMR difference everywhere.

27. **Comments from the referee:** Page 5, lines 2 and 3 (counting from -2): At face value, the 30-day smoothing and 4-day filtering appear to be contradictory. If the 30 data points are for 30 days worth of observations, then surely such a smoothing is going to filter far more aggressively than 4 days? Are there more than 30 points per day? Is this related to the issue of having more multiple MLS matches to a single GROMOS measurement? If so, this needs to be made much clearer. Plus, the impact of this smoothing is going to vary quite significantly depending on how many points there are on a given day. Why not simply smooth on a daily rather than a point-by-point basis (average of all differences within an n-day window)? Again, all this needs to be much more clearly described.

    **Author's response:**
    As we have changed the spatial criteria of coincidence the number of coincident profiles has changed as well. Therefore, now we have performed moving average over 7 points which corresponds to around 1 week. Performing a daily smoothing in the time series will produce noisy and unclear Figures and hence difficulties to interpret them.

    **Author's changes in the manuscript:** Page 5, lines 2 and 3:  All time series have been smoothed by a moving average over 7 points (∼1week).

[Figure]

Figure 4: Time series of averaged daytime and nighttime $O_3$ VMR measurements of GROMOS (blue line) and MLS (red line) for the period from July 2009 to November 2016 at different pressure levels. An averaging kernel smoothing has been applied to the series of the MLS measurements coincident in time and space with the GROMOS measurements. Both time series are smoothed over 7 points or 1 week in time by a moving average

28. **Comments from the referee:** Page 5, line 8: "almost perfect" is very much in the eye of the beholder, and in my eye your scatter plots are far from it. To me "almost perfect" is at the $> 0.999$ level of correlation, where the points are all but indistinguishable from the 1:1 line, with perhaps just one or two strays. I suggest you use more measured language.

    **Author's response:**
    No comment.

    **Author's changes in the manuscript:** Page 5, line 8:

29. **Comments from the referee:** Page 5, line 9: Odd way to phrase it, simply say that the black line is close to the green one to one line.

    **Author's response:**
    We agree with the referee.

    **Author's changes in the manuscript:** Page 5, line 8–9: The black lines, linear regression lines of the observations, are close to the green one to one lines, $O_3(MLS)=O_3(GROMOS)$.

30. **Comments from the referee:** Page 5, line 21: "variation is also expected"

    **Author's response:**
    Thanks for spotting. We have corrected this.

    **Author's changes in the manuscript:** Page 5, line 21: ... therefore an annual variation is also expected

31. **Comments from the referee:** Page 6, lines -2 to 2: As discussed above, more discussion is needed here. Some more investigation is needed as to why the amplitudes of the cycles are so different. You don't even tell us if we should be surprised by this level of disagreement. Note that the MLS averaging kernels imply not insignificant vertical smoothing at these altitudes for this instrument too. When taken in conjunction with the possible latitudinal gradient, are there plausible reasons to explain the differences based on sampling etc. alone, or is the only feasible explanation some instrumental/calibration difference? If nothing else, raise these questions and identify a route to answering them. Could the diurnal cycle in temperature (and thus the pressure/height relationship) play any role in this (from a measurement characteristics point of view rather than an atmospheric science one)? This manuscript would greatly benefit from an analysis, or at least an identification, of all the potential factors involved.

    **Author's response:**
    According to Sonnemann et al. (2007), *the MMM is an effect occurring at high latitudes close to the polar night terminator around 72 km altitude during nighttime in the winter half of the year and extends into middle latitudes with decreasing amplitude.* The observed sharp decrease of the amplitude of the MMM of ozone is due to the strong latitudinal gradient between high and middle latitudes. In fact, it is surprising that we can observe the effect of MMM at our latitude. Therefore, the difference in latitude between Lindau and Bern may have such impact in the amplitudes of the annual variability of mesospheric ozone due to the MMM. However it could also be due to some other effects like for example, differences in

the retrieval algorithms between Bern and Lindau, different instruments used to perform the measurements, different calculation methods...

**Author's changes in the manuscript:** Page 6, line 4: Nevertheless, our results are the expected since this maximum of mesospheric ozone during nighttime in winter is related to the middle mesospheric maximum of ozone (MMM) and according to Sonnemann et al. (2007) its effect extends into midlatitudes with decreasing amplitude.

32. **Comments from the referee:** Page 6, lines 13-15: This discussion is unclear, at least to me. If the orange points are smoothed by 10 points, is that 10 days? How does this number related to the ~7 years between 2009 and 2016. I don't get how the 10-point and 30-point smoothings are related.

    **Author's response:**
    We have repeated the comparison by changing the spatial criteria of coincidence and now the number of coincident profiles has changed. In the first panel of Figure 7 (former Figure 6) the moving average is over 30 points, roughly 1 month and in the second panel we used a moving average over 7 points which corresponds to around 1 week. The purpose of the smoothing is to help the interpretation of the results.

    **Author's changes in the manuscript:** Page 5, line 30: All time series displayed in both panels of Figure 6 have been smoothed in time by a moving average over 15 data points (~1 week).
    Page 6, lines 13-15: ...under assessment. Both time series were smoothed in time by a moving average over 30 points ($\sim$ 1 month). ... the second panel of Figure 7 show a moving average over 7 data points (1 week) with the aim to clarify the understanding of Figure 7.

33. **Comments from the referee:** Figures: In general, all the figures use overly heavy line thicknesses. While it may be OK for the lines themselves (though rather on the heavy side), the line width used is far to heavy for the axes. Also the font should be slightly (~20-50%) larger, and perhaps not bold, for greater clarity.
    Figure 2: Suggest "mean absolute difference" → "mean mixing ratio difference". Also, how is 'its uncertainty" (last line) defined? Do you mean standard deviation?

    **Author's response:**
    We have calculated the mean relative difference profile and the VMR difference profile separating daytime and nighttime values, accordingly Figure 3 (former Figure 2) has changed.

    **Author's changes in the manuscript:** Figure 3

[revised manuscript text omitted]

---

## Referee Report (RR1)

- I think that this revised manuscript is an improvement, although some things are still not all that clear in terms of the differences between the ground-based and the satellite data, i.e. where the truth lies when the differences in VMR or the day-to-night ratios become larger (and this is difficult to assess, I realize). It is also therefore not obvious that the newer retrieval leads to better results, in terms of biases at least.

- I view this revised version as suitable for publication after some minor/editorial changes, see the suggestions below.

Minor/editorial comments

- Page 1, Line 9 (P1, L1), not clear what "over 20%" really means (larger or better/less than 20%, or about 20%?).
  > also, L15, I would reword as "related to the ozone middle mesospheric maximum (MMM)."

- P2, L14/15, "performaed an analysis of the ozone diurnal variation and its amplitude (night-to-day ratio) in the middle mesosphere at 0.05 hPa (70 km)." [although since the study was also performed at other altitudes (no?), maybe mentioning just that altitude is not needed].
  > L20, I would delete "the" in front of "nighttime".

- P4, L8, I would delete "of the" before "ECMWF".
  > L22, I suggest "decrease exponentially with a correlation...".

- P5, L2, I would say "and in the case of version 2021 it is 30 minutes."
  > L6, I would say "standard deviations"
  > L8, I suggest changing "are ranging" to "range".
  > L14, "The left panel shows".
  > L32, change "limit" to "limits".
  > L33, For technical details and measurement principle...

- P7, L12, For an overview of the differences ...

- P8, L7, I suggest deleting "for the already mentioned time period"
  > L9, in order to provide an insight into the observability...